# Provable Online CP/PARAFAC Decomposition of a Structured Tensor via Dictionary Learning

**Sirisha Rambhatla** [1]
sirishar@usc.edu

**Xingguo Li** [2]
xingguol@cs.princeton.edu

**Jarvis Haupt** [3]
jdhaupt@umn.edu

[1] Computer Science Department, University of Southern California
[2] Computer Science Department, Princeton University
[3] Department of Electrical and Computer Engineering, University of Minnesota – Twin Cities

## Abstract

We consider the problem of factorizing a structured 3-way tensor into its constituent Canonical Polyadic (CP) factors. This decomposition, which can be viewed as a generalization of singular value decomposition (SVD) for tensors, reveals how the tensor dimensions (features) interact with each other. However, since the factors are *a priori* unknown, the corresponding optimization problems are inherently non-convex. The existing guaranteed algorithms which handle this non-convexity incur an irreducible error (bias), and only apply to cases where all factors have the same structure. To this end, we develop a provable algorithm for online structured tensor factorization, wherein one of the factors obeys some incoherence conditions, and the others are sparse. Specifically we show that, under some relatively mild conditions on initialization, rank, and sparsity, our algorithm recovers the factors *exactly* (up to scaling and permutation) at a linear rate. Complementary to our theoretical results, our synthetic and real-world data evaluations showcase superior performance compared to related techniques.

## 1 Introduction

Canonical Polyadic (CP) /PARAFAC decomposition aims to express a tensor as a sum of rank-1 tensors, each of which is formed by the outer-product (denoted by "∘") of constituent factors columns. In this work, we consider the *online* factorization of a structured tensor 3-way tensor $\underline{\mathbf{Z}}^{(t)} \in \mathbb{R}^{n \times J \times K}$ arriving at time $t$, as

$$\underline{\mathbf{Z}}^{(t)} = \sum_{i=1}^{m} \mathbf{A}_i^* \circ \mathbf{B}_i^{*(t)} \circ \mathbf{C}_i^{*(t)} = [\![\mathbf{A}^*, \mathbf{B}^{*(t)}, \mathbf{C}^{*(t)}]\!], \tag{1}$$

where $\mathbf{A}_i^*$, $\circ\mathbf{B}_i^{*(t)}$ and $\mathbf{C}_i^{*(t)}$ are columns of factors $\mathbf{A}^*$, $\mathbf{B}^{*(t)}$, and $\mathbf{C}^{*(t)}$, respectively, and are *a priori* unknown. A popular choice for the batch setting (not online) is via the alternating least squares (ALS) algorithm, where appropriate regularization terms (such as $\ell_1$ loss for sparsity) are added to the least-square objective to steer towards specific solutions [1–4]. However, these approaches suffer from three major issues – a) the non-convexity of associated formulations makes it challenging to establish recovery and convergence guarantees, b) one may need to solve an implicit model selection problem (e.g., choose the tensor rank $m$), and c) regularization may be computationally expensive, and may not scale well in practice.

Recent works for guaranteed tensor factorization – based on tensor power method [5], convex relaxations [6], sum-of-squares formulations [7–9], and variants of ALS algorithm [10] – have focused on recovery of tensor factors wherein all factors have a common structure; based on some notion of incoherence of individual factor matrices such as sparsity, incoherence, or both [11]. Furthermore, these algorithms a) incur *bias* in estimation, b) are computationally expensive in practice, and c) are not amenable for online (streaming) tensor factorization; See Table 1. Consequently, there is a need to develop fast, scalable provable algorithms for exact (unbiased) factorization of structured tensors arriving (or processed) in a streaming fashion (online), generated by heterogeneously structured factors. To this end, we develop

a provable algorithm to recover the unknown factors of tensor(s) $\underline{\mathbf{Z}}^{(t)}$ in Fig.1 (arriving, or made available for sequential processing, at an instance $t$), assumed to be generated as (1), wherein the factor $\mathbf{A}^*$ is *incoherent* and fixed (deterministic), and the factors $\mathbf{B}^{*(t)}$ and $\mathbf{C}^{*(t)}$ are sparse and vary with $t$ (obey some randomness assumptions).

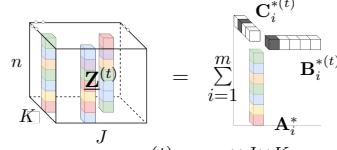

**Figure 1:** Tensor $\underline{\mathbf{Z}}^{(t)} \in \mathbb{R}^{n \times J \times K}$ of interest, a few mode-1 fibers are dense.

**Model Justification**: The tensor factorization task of interest arises in streaming applications where users interact only with a few items at each time $t$, i.e. the user-item interactions are *sparse*. Here, the fixed incoherent factor $\mathbf{A}^*$ columns model the underlying fixed interactions patterns (*signatures*). At time $t$, a fresh observation tensor $\underline{\mathbf{Z}}^{(t)}$ arrives, and the task is to estimate sparse factors (users and items), and the incoherent factor (patterns). This estimation procedure reveals users $\mathbf{B}_i^*$ and items $\mathbf{C}_i^*$ sharing the same pattern $\mathbf{A}_i^*$, i.e. the underlying clustering, and finds applications in scrolling pattern analysis in web analytics [12], sports analytics (section 5.2), patient response to probes [13, 14], electro-dermal response to audio-visual stimuli [15, 16], and organizational behavior via email activity [1, 17].

## 1.1 Overview of the results

We take a matrix factorization view of the tensor factorization task to develop an online provable tensor factorization algorithm for exact recovery of the constituent factors. Leveraging the structure, we envision the non-zero fibers as being generated by a dictionary learning model, where the data samples $\mathbf{y}_{(j)} \in \mathbb{R}^n$ are assumed to be generated as follows from an *a priori* unknown dictionary $\mathbf{A}^* \in \mathbb{R}^{n \times m}$ and sparse coefficients $\mathbf{x}_{(j)}^* \in \mathbb{R}^m$.

$$\mathbf{y}_{(j)} = \mathbf{A}^* \mathbf{x}_{(j)}^*, \ \|\mathbf{x}_{(j)}^*\|_0 \leq s \ \text{ for all } \ j = 1, 2, \dots \tag{2}$$

This modeling procedure includes a matricization or *flattening* of the tensor, which leads to a Kronecker (Khatri-Rao) dependence structure among the elements of the resulting coefficient matrix; see section 4. As a result, the main challenges in developing recovery guarantees are to: a) analyze the Khatri Rao product (KRP) structure to identify and quantify data samples (non-zero fibers) available for learning, b) establish guarantees on the resulting sparsity structure, and c) develop a SVD-based guaranteed algorithm to successfully untangle the sparse factors using corresponding coefficient matrix estimate and the underlying KRP structure. Also, our matricization-based analysis can be of independent interest.

## 1.2 Contributions

We develop an algorithm to recover the CP factors of tensor(s) $\underline{\mathbf{Z}}^{(t)} \in \mathbb{R}^{n \times J \times K}$, arriving (or made available) at time $t$, generated as per (1) from constituent factors $\mathbf{A}^* \in \mathbb{R}^{n \times m}$, $\mathbf{B}^{*(t)} \in \mathbb{R}^{J \times m}$, and $\mathbf{C}^{*(t)} \in \mathbb{R}^{K \times m}$, where the unit-norm columns of $\mathbf{A}^*$ obey some incoherence assumptions, and $\mathbf{B}^{*(t)}$ and $\mathbf{C}^{*(t)}$ are sparse. Our specific contributions are:

- **Exact recovery and linear convergence**: Our algorithm `TensorNOODL`, to the best of our knowledge, is the first to accomplish recovery of the true CP factors of this structured tensor(s) $\underline{\mathbf{Z}}^{(t)}$ *exactly* (up to scaling and permutations) at a linear rate. Specifically, starting with an appropriate initialization $\mathbf{A}^{(0)}$ of $\mathbf{A}^*$, we have $\mathbf{A}_i^{(t)} \to \mathbf{A}_i^*$, $\widehat{\mathbf{B}}_i^{(t)} \to \pi_{B_i} \mathbf{B}_i^{*(t)}$, and $\widehat{\mathbf{C}}_i^{(t)} \to \pi_{C_i} \mathbf{C}_i^{*(t)}$, as iterations $t \to \infty$, for constants $\pi_{B_i}$ and $\pi_{C_i}$.
- **Provable algorithm for heterogeneously-structured tensor factorization**: We consider the *exact* tensor factorization (an inherently non-convex task) when the factors do not obey same structural assumptions. That is, our algorithmic procedure overcomes the non-convexity bottleneck suffered by related optimization-based ALS formulations.
- **Online, fast, and scalable**: The online nature of our algorithm, separability of updates due to bio-inspired *neural plausibility*, and relatively easy to tune parameters, make it suitable for large-scale distributed implementations. Furthermore, our numerical simulations (both synthetic and real-world) demonstrate superior performance in terms of accuracy, number of iterations, demonstrating its applicability to real-world tasks.

Furthermore, although estimating the rank of a given tensor is NP hard, the incoherence assumption on $\mathbf{A}^*$, and distributional assumptions on $\mathbf{B}^{*(t)}$ and $\mathbf{C}^{*(t)}$, ensure that our matrix factorization view is *rank revealing* [18]. In other words, our assumptions ensure that the dictionary initialization algorithms (such as [19]) can recover the rank of the tensor. Following this, `TensorNOODL` recovers the true factors (up to scaling and permutation) whp.

**Table 1:** Comparing provable algorithms for tensor factorization and dictionary learning. As shown here, the existing provable techniques do not apply where $\mathbf{A}$: incoherent, $(\mathbf{B}, \mathbf{C})$: sparse.

| Method | Conditions | | | Recovery Guarantees | |
|---|---|---|---|---|---|
| | Model Considered | Rank | Initialization Constraints | Estimation Bias | Convergence |
| TensorNOODL (this work) | $\mathbf{A}$: incoherent, $(\mathbf{B}, \mathbf{C})$: sparse | $m = \mathcal{O}(n)$ | $\mathcal{O}^*\left(\frac{1}{\log(n)}\right)$ | No Bias | Linear |
| Sun et al. [11]‡ | $(\mathbf{A}, \mathbf{B}, \mathbf{C})$: all incoherent and sparse | $m = o(n^{1.5})$ | $o(1)$ | $\|\mathbf{A}_{ij} - \widehat{\mathbf{A}}_{ij}\|_\infty = \mathcal{O}(\frac{1}{n^{0.25}})^\dagger$ | Not established |
| Sharan and Valiant [10]‡ | $(\mathbf{A}, \mathbf{B}, \mathbf{C})$: all incoherent | $m = o(n^{0.25})$ | Random | $\|\mathbf{A}_i - \widehat{\mathbf{A}}_i\|_2 = \mathcal{O}(\sqrt{\frac{m}{n}})^\dagger$ | Quadratic |
| Anandkumar et al. [5]‡ | $(\mathbf{A}, \mathbf{B}, \mathbf{C})$: all incoherent | $m = \mathcal{O}(n)$ | $\mathcal{O}^*\left(\frac{1}{\sqrt{n}}\right)^\P$ | $\|\mathbf{A}_i - \widehat{\mathbf{A}}_i\|_2 = \widetilde{\mathcal{O}}(\frac{1}{\sqrt{n}})^\dagger$ | Linear§ |
| | | $m = o(n^{1.5})$ | $\mathcal{O}(1)$ | $\|\mathbf{A}_i - \widehat{\mathbf{A}}_i\|_2 = \widetilde{\mathcal{O}}(\frac{\sqrt{m}}{n})^\dagger$ | Linear |
| Arora et al. [19] | Dictionary Learning (2) | $m = \mathcal{O}(n)$ | $\mathcal{O}^*\left(\frac{1}{\log(n)}\right)$ | $\mathcal{O}(\sqrt{s/n})$ | Linear |
| | | $m = \mathcal{O}(n)$ | $\mathcal{O}^*\left(\frac{1}{\log(n)}\right)$ | *Negligible* bias § | Linear |
| Mairal et al. [20] | Dictionary Learning (2) | Convergence to stationary point; similar guarantees by Huang et al. [21]. | | | |

‡ This procedure is not *online*. † Result applies for each $i \in [1, m]$. ¶ Polynomial number of initializations $m^{\beta^2}$ are required, for $\beta \geq m/n$. § The procedure has an *almost* Quadratic rate initially.

## 1.3 Related works

**Tensor Factorization**: Canonical polyadic (CP)/PARAFAC decomposition (1) captures relationships between the latent factors, where the number of rank-1 tensors define the rank for a tensor. Unlike matrix decompositions, tensor factorizations can be unique under relatively mild conditions [22, 23]. However, determining tensor rank is NP-hard [24], and so are tasks like tensor decompositions [25]. Nevertheless, regularized ALS-based approaches emerged as a popular choice to impose structure on the factors, however establishing convergence to even a stationary point is difficult [26]; see also [27]. The variants of ALS with some convergence guarantees do so at the expense of complexity [28, 29], and convergence rate [30]; See also [1] and [18]. On the other hand, guaranteed methods for tensor factorization initially relied on a computationally expensive orthogonalizing step (*whitening*), and therefore, did not extend to the overcomplete setting ($m > n$) [31–37]. As a result, works such as [5, 6, 38], relaxed orthogonality to an *incoherence* condition to handle the overcomplete setting. To counter the complexity of these methods, [10] developed a provable ALS variant using orthogonalization, however, this precludes its use in overcomplete settings.

**Dictionary Learning**: We now provide a brief overview of the dictionary learning literature. Popularized by the rich sparse inference literature, *overcomplete* ($m \geq n$) representations lead to sparse(r) representations which are robust to noise; see [39–41]. Learning such sparsifying overcomplete representations is known as *dictionary learning* [20, 42–45]. Analogous to the ALS algorithm, the alternating minimization-based techniques became widely popular in practice, however theoretical guarantees were still limited. Provable algorithms for under- and over-complete settings were developed, however their computational complexity and initialization requirements limited their use [7, 46–48]. Tensor factorization algorithms have also been used to learn orthogonal ([7] and [8]), and convolutional [35] dictionaries. More recently, [49] proposed `NOODL`: a simple, scalable gradient descent-based algorithm for joint estimation of the dictionary and the coefficients, for *exact* recovery of both factors at a linear rate. Although this serves as a great starting point, tensor factorization task cannot be handled by a mere "lifting" due to the induced dependence structure.

Overall, the existing provable techniques (Table 1) in addition to being computationally expensive, incur an irreducible error (bias) in estimation and apply to cases where all factors obey the same conditions. Consequently, there is a need for fast and scalable provable tensor factorization techniques which can recover structured factors with no estimation bias.

**Notation.** Bold, lower-case ($\mathbf{v}$) and upper-case ($\mathbf{M}$) letters, denote vectors and matrices, respectively. We use $\mathbf{M}_i$, $\mathbf{M}_{(i,:)}$, $\mathbf{M}_{ij}$ (also $\mathbf{M}(i,j)$), and $\mathbf{v}_i$ (also $\mathbf{v}(i)$) to denote the $i$-th column, $i$-th row, $(i, j)$ element, respectively. We use "$\odot$" and "$\otimes$" to denote the Khatri-Rao (column-wise Kronecker product) and Kronecker product, respectively. Next, we use $(\cdot)^{(n)}$ to denote the $n$-th iterate, and $(\cdot)_{(n)}$ for the $n$-th data sample. We also use standard Landau notations $\mathcal{O}(\cdot), \Omega(\cdot)$ ($\widetilde{\mathcal{O}}(\cdot), \widetilde{\Omega}(\cdot)$) to denote the asymptotic behavior (ignoring log factors). Also, for a constant $L$ (independent of $n$), we use $g(n) = \mathcal{O}^*(f(n))$ to indicate that $g(n) \leq Lf(n)$. We use $c(\cdot)$ for constants determined by the quantities in $(\cdot)$. Also, we define $\mathcal{T}_\tau(z) := z \cdot \mathbb{1}_{|z| \geq \tau}$ as the hard-thresholding operator, where "$\mathbb{1}$" is the indicator function, and $\text{supp}(\cdot)$ for the support (set of non-zero elements) and $\text{sign}(\cdot)$ for element-wise sign. Also, $(.)^{(r)}$ denotes potential iteration dependent parameters. See Appendix A.

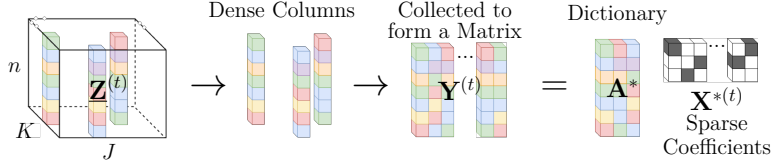

**Figure 2:** Problem Formulation: The dense columns of $\mathbf{Z}^{(t)} \in \mathbb{R}^{n \times J \times K}$ are collected in a matrix $\mathbf{Y}^{(t)}$. Then $\mathbf{Y}^{(t)}$ is viewed as arising from a dictionary learning model.

## 2    Problem Formulation

Our formulation is shown in Fig. 2. Here, our aim is to recover the CP factors of tensors $\{\underline{\mathbf{Z}}^{(t)}\}_{t=0}^{T-1}$ assumed to be generated at each iteration as per (1). Without loss of generality, let the factor $\mathbf{A}^*$ follow some incoherence assumptions, while the factors $\mathbf{B}^{*(t)}$ and $\mathbf{C}^{*(t)}$ be sparse. Now, the *mode*-1 *unfolding* or matricization $\mathbf{Z}_1^{(t)} \in \mathbb{R}^{JK \times n}$ of $\underline{\mathbf{Z}}^{(t)}$ is given by

$$\mathbf{Z}_1^{(t)\top} = \mathbf{A}^*(\mathbf{C}^{*(t)} \odot \mathbf{B}^{*(t)})^\top = \mathbf{A}^* \mathbf{S}^{*(t)}, \tag{3}$$

where $\mathbf{S}^{*(t)} \in \mathbb{R}^{m \times JK}$ is $\mathbf{S}^{*(t)} := (\mathbf{C}^{*(t)} \odot \mathbf{B}^{*(\mathbf{t})})^\top$. As a result, matrix $\mathbf{S}^{*(t)}$ has a *transposed Khatri-Rao* structure, i.e. the $i$-th row of $\mathbf{S}^{*(t)}$ is given by $(\mathbf{C}_i^{*(t)} \otimes \mathbf{B}_i^{*(t)})^\top$. Further, since $\mathbf{B}^{*(t)}$ and $\mathbf{C}^{*(t)}$ are sparse, only a few $\mathbf{S}^{*(t)}$ columns (say $p$) have non-zero elements. Now, let $\mathbf{Y}^{(t)} \in \mathbb{R}^{n \times p}$ be a matrix formed by collecting the non-zero $\mathbf{Z}_1^{(t)\top}$ columns, we have

$$\mathbf{Y}^{(t)} = \mathbf{A}^* \mathbf{X}^{*(t)}, \tag{4}$$

where $\mathbf{X}^{*(t)} \in \mathbb{R}^{m \times p}$ denotes the sparse matrix corresponding to the non-zero columns of $\mathbf{S}^{*(t)}$. Since recovering $\mathbf{A}^*$ and $\mathbf{X}^{*(t)}$ given $\mathbf{Y}^{(t)}$ is a dictionary learning task (2), we can now employ a dictionary learning algorithm (such as `NOODL`) which *exactly* recovers $\mathbf{A}^*$ (the *dictionary*) and $\mathbf{X}^{*(t)}$ (the *sparse coefficients*) at each time step $t$ of the (online) algorithm. The exact recovery of $\mathbf{X}^{*(t)}$ enables recovery of $\mathbf{B}^{*(t)}$ and $\mathbf{C}^{*(t)}$ using our untangling procedure.

## 3    Algorithm

We begin by presenting the algorithmic details referring to relevant assumptions, we then analyze the model assumptions and the main result in section 4. `TensorNOODL` (Alg. 1) operates by casting the tensor decomposition problem as a dictionary learning task. Initially, Alg. 1 is given a $(\epsilon_0, 2)$-close (defined below) estimate $\mathbf{A}^{(0)}$ of $\mathbf{A}^*$ for $\epsilon_0 = \mathcal{O}^*(1/\log(n))$. This initialization, which can be achieved by algorithms such as Arora et al. [19], ensures that the estimate $\mathbf{A}^{(0)}$ is both, column-wise and in spectral norm sense, close to $\mathbf{A}^*$.

**Definition 1 $((\epsilon, \kappa)$-closeness)** *A matrix $\mathbf{A}$ is $(\epsilon, \kappa)$-close to $\mathbf{A}^*$ if $\|\mathbf{A} - \mathbf{A}^*\| \leq \kappa \|\mathbf{A}^*\|$, and if there is a permutation $\pi : [m] \to [m]$ and a collection of signs $\sigma : [m] \to \{\pm 1\}$ such that $\|\sigma(i)\mathbf{A}_{\pi(i)} - \mathbf{A}_i^*\| \leq \epsilon, \ \forall \ i \in [m].$*

Next, we sequentially provide the tensors to be factorized, $\{\underline{\mathbf{Z}}^{(t)}\}_{t=0}^{T-1}$ (generated independently as per (1)) at each iteration $t$. The algorithm proceeds in the following stages.

**I. Estimate Sparse Matrix $\mathbf{X}^{*(t)}$:** We use $R$ iterative hard thresholding (IHT) steps (6) – with step-size $\eta_x^{(r)}$ and threshold $\tau^{(r)}$ chosen according to **A.6** – to arrive at an estimate $\widehat{\mathbf{X}}^{(t)}$ (or $\mathbf{X}^{(R)(t)}$). Iterations $R$ are determined by the target tolerance $(\delta_R)$ of the desired coefficient estimate, i.e. we choose $R = \Omega(\log(1/\delta_R))$, where $(1 - \eta_x^{(r)})^R \leq \delta_R$.

**II. Estimate $\mathbf{B}^*$ and $\mathbf{C}^*$:** As discussed in section 2, the tensor matricization leads to a Khatri-Rao dependence structure between the factors $\mathbf{B}^{*(t)}$ and $\mathbf{C}^{*(t)}$. To recover these, we develop a SVD-based algorithm (Alg. 2) to estimate sparse factors ($\mathbf{B}^{*(t)}$ and $\mathbf{C}^{*(t)}$) using an element-wise $\zeta$-close estimate of $\mathbf{S}^{*(t)}$, i.e., $|\widehat{\mathbf{S}}_{ij}^{(t)} - \mathbf{S}_{ij}^{*(t)}| \leq \zeta$. Here, we form the estimate $\widehat{\mathbf{S}}^{(t)}$ of $\mathbf{S}^{*(t)}$ by placing columns of $\widehat{\mathbf{X}}^{(t)}$ at their corresponding locations of $\mathbf{Z}_1^{(t)\top}$ to the Khatri-Rao structure (`TensorNOODL` is agnostic to the tensor structure of the data since it only operates on the non-zero fibers $\mathbf{Y}^{(t)}$ of $\mathbf{Z}_1^{(t)\top}$; see (4) and Fig. 2). Our recovery result for $\widehat{\mathbf{X}}^{(t)}$ guarantees that $\widehat{\mathbf{S}}^{(t)}$ has the same sign and support as $\widehat{\mathbf{S}}^{*(t)}$, we therefore provably recover the original Khatri-Rao product structure.

**III. Update $\mathbf{A}^*$ estimate** : We use $\mathbf{X}^{*(t)}$ estimate to update $\mathbf{A}^{(t)}$ by an approximate gradient descent strategy (8) with step size $\eta_A$ (**A.5**). The algorithm requires $T = \max(\Omega(\log(1/\epsilon_T)), \Omega(\log(\sqrt{s}/\delta_T)))$ for $\|\mathbf{A}_i^{(T)} - \mathbf{A}_i^*\| \le \epsilon_T, \forall i \in [m]$ and $|\widehat{\mathbf{X}}_{ij}^{(T)} - \mathbf{X}_{ij}^{*(t)}| \le \delta_T$.

**Runtime**: The runtime of `TensorNOODL` is $\mathcal{O}(mnp \log(\frac{1}{\delta_R}) \max(\log(\frac{1}{\epsilon_T}), \log(\frac{\sqrt{s}}{\delta_T}))$ for $p = \Omega(ms^2)$. Furthermore, since $\mathbf{X}^*$ columns can be estimated independently in parallel, `TensorNOODL` is scalable and can be implemented in highly distributed settings.

## 4  Main Result

We now formalize our model assumptions and state our main result; details in Appendix B.

**Model Assumptions**: First, we require that $\mathbf{A}^*$ is $\mu$-incoherent (defined below), which defines the notion of incoherence for $\mathbf{A}^*$ columns (refered to as *dictionary*).

**Definition 2**  *A matrix $\mathbf{A} \in \mathbb{R}^{n \times m}$ with unit-norm columns is $\mu$-incoherent if for all $i \ne j$ the inner-product between the columns of the matrix follow $|\langle \mathbf{A}_i, \mathbf{A}_j \rangle| \le \mu/\sqrt{n}$.*

This ensures that dictionary columns are distinguishable, akin to relaxing the orthogonality constraint. Next, we assume that sparse factors $\mathbf{B}^{*(t)}$ and $\mathbf{C}^{*(t)}$ are drawn from distribution classes $\Gamma_{\alpha,C}^{\mathrm{sG}}$ and $\Gamma_{\beta}^{\mathrm{Rad}}$, respectively, here $\Gamma_{\gamma,C}^{\mathrm{sG}}$ and $\Gamma_{\gamma}^{\mathrm{Rad}}$ are defined as follows.

---

**Algorithm 1** TENSORNOODL: Neurally plausible alternating Optimization-based Online Dictionary Learning for Tensor decompositions.

---

**Input**: Structured tensor $\underline{\mathbf{Z}}^{(t)} \in \mathbb{R}^{n \times J \times K}$ at each $t$ generated as per (1). Parameters $\eta_A$, $\eta_x$, $\tau$, $T$, $C$, and $R$ as per **A.3**, **A.5**, and **A.6**.

**Output**: Dictionary $\mathbf{A}^{(t)}$ and the factor estimates $\mathbf{B}^{(t)}$ and $\mathbf{C}^{(t)}$ (corresponding to $\underline{\mathbf{Z}}^{(t)}$) at $t$.

**Initialize**: Estimate $\mathbf{A}^{(0)}$, which is $(\epsilon_0, 2)$-near to $\mathbf{A}^*$ for $\epsilon_0 = \mathcal{O}^*(1/\log(n))$; see Def. 1.

**for** $t = 0$ **to** $T - 1$ **do**

  **I. Estimate Sparse Matrix $\mathbf{X}^{*(t)}$:**

  **Initialize:** $\mathbf{X}^{(0)(t)} = \mathcal{T}_{C/2}(\mathbf{A}^{(t)\top}\mathbf{Y}^{(t)})$    See Def.3 (5)

  **for** $r = 0$ **to** $R - 1$ **do**

$$\mathbf{X}^{(r+1)(t)} = \mathcal{T}_{\tau^{(r)}}(\mathbf{X}^{(r)(t)} - \eta_x^{(r)}\mathbf{A}^{(t)\top}(\mathbf{A}^{(t)}\mathbf{X}^{(r)} - \mathbf{Y}^{(t)}))$$
                          (6)

  **end**

  $\widehat{\mathbf{X}}^{(t)} := \mathbf{X}^{(R)(t)}$.

  **II. Recover Sparse Factors $\mathbf{B}^*$ and $\mathbf{C}^*$:**

  Form $\widehat{\mathbf{S}}^{(t)}$ by putting back columns of $\widehat{\mathbf{X}}^{(t)}$ at the non-zero column locations of $\mathbf{Z}_1^{(t)\top}$.

  $[\widehat{\mathbf{B}}^{(t)}, \widehat{\mathbf{C}}^{(t)}] = $ UNTANGLE-KRP$(\widehat{\mathbf{S}}^{(t)})$

  **III. Update Dictionary Factor $\mathbf{A}^{(t)}$:**

  $\widehat{\mathbf{g}}^{(t)} = \frac{1}{p}(\mathbf{A}^{(t)}\widehat{\mathbf{X}}_{\mathrm{indep}}^{(t)} - \mathbf{Y}^{*(t)})\mathrm{sign}(\widehat{\mathbf{X}}_{\mathrm{indep}}^{(t)})^\top$   (7)

  $\mathbf{A}^{(t+1)} = \mathbf{A}^{(t)} - \eta_A\,\widehat{\mathbf{g}}^{(t)}$         (8)

  $\mathbf{A}_i^{(t+1)} = \mathbf{A}_i^{(t+1)}/\|\mathbf{A}_i^{(t+1)}\| \; \forall \; i \in [m]$

**end**

---

**Algorithm 2** UNTANGLE KHATRI-RAO PRODUCT (KRP): Recovering the Sparse factors

---

**Input**: Estimate $\widehat{\mathbf{S}}^{(t)}$ of the KRP $\mathbf{S}^{*(t)}$

**Output**: Estimates $\widehat{\mathbf{B}}^{(t)}$ and $\widehat{\mathbf{C}}^{(t)}$ of $\mathbf{B}^{*(t)}$ and $\mathbf{C}^{*(t)}$.

**for** $i = 1 \ldots m$ **do**

  **Reshape:** $i$-th row of $\widehat{\mathbf{S}}^{(t)}$ into $\mathbf{M}^{(i)} \in \mathbb{R}^{J \times K}$.

  **Set:** $\widehat{\mathbf{B}}_i^{(t)} \leftarrow \sqrt{\sigma_1}\mathbf{u}_1$, and $\widehat{\mathbf{C}}_i^{(t)} \leftarrow \sqrt{\sigma_1}\mathbf{v}_1$, where $\sigma_1$, $\mathbf{u}_1$, and $\mathbf{v}_1$ are the principal left and right singular vectors of $\mathbf{M}^{(i)}$, respectively.

**end**

---

**Definition 3 (Distribution Class $\Gamma_{\gamma,C}^{\mathrm{sG}}$ and $\Gamma_{\gamma}^{\mathrm{Rad}}$)** *A matrix $\mathbf{M}$ belongs to class*

- $\Gamma_{\gamma}^{\mathrm{Rad}}$: *if each entry of $\mathbf{M}$ is independently non-zero with probability $\gamma$, and the values at the non-zero locations are drawn from the Rademacher distribution.*

- $\Gamma_{\gamma,C}^{\mathrm{sG}}$: *if each entry of $\mathbf{M}$ is independently non-zero with probability $\gamma$, and the values at the non-zero locations are sub-Gaussian, zero-mean with unit variance and bounded away from $C$ for some positive constant $C \le 1$, i.e., $|\mathbf{M}_{ij}| \ge C$ for $(i, j) \in \mathrm{supp}(\mathbf{M})$.*

In essence, we assume that elements of $\mathbf{B}^{*(t)}$ ($\mathbf{C}^{*(t)}$) are non-zero with probability $\alpha$ ($\beta$), and that for $\mathbf{B}^{*(t)}$ the values at the non-zero locations are drawn from a zero-mean unit-variance sub-Gaussian distribution, bounded away from zero, and the non-zero values of $\mathbf{C}^{*(t)}$ are drawn from the Rademacher distribution [2].

**Analyzing the Khatri-Rao Dependence**: We now turn our attention to the KR dependence structure of $\mathbf{S}^{*(t)}$. Fig. 3 shows a row of the matrix $\mathbf{S}^{*(t)}$, each entry of which is formed by multiplication of an element of $\mathbf{C}_i^{*(t)}$ with each element of columns of $\mathbf{B}_i^{*(t)}$. Consequently, each row of the resulting matrix $\mathbf{S}^{*(t)}$ has $K$ *blocks* (of size $J$), where the $k$-th block is controlled by $\mathbf{C}_{k,i}^{*(t)}$, and therefore the $(i, j)$-th entry of $\mathbf{S}^{*(t)}$ can be written as

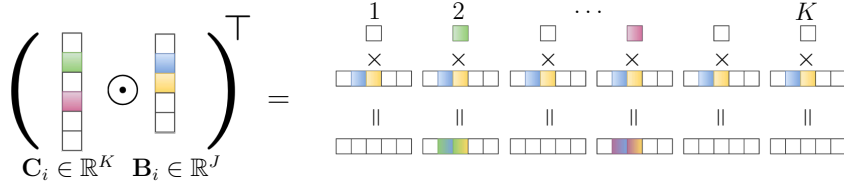

**Figure 3:** Transposed Khatri-Rao dependence.

$$\mathbf{S}_{ij}^{*(t)} = \mathbf{C}^{*(t)}\big(\big\lfloor \tfrac{j}{J} \big\rfloor + 1, i\big)\, \mathbf{B}^{*(t)}\big(j - J\big\lfloor \tfrac{j}{J} \big\rfloor, i\big). \tag{9}$$

Depending upon $\alpha(\beta)$, $\mathbf{S}^{*(t)}$ (consequently $\mathbf{Z}_1^{(t)\top}$) may have all-zero (degenerate) columns, therefore, we only use non-zero columns $\mathbf{Y}^{(t)}$ of $\mathbf{Z}_1^{(t)\top}$. Next, although elements in a column of $\mathbf{S}^{*(t)}$ are independent, the KR structure induces a dependence in a row when they depend on the same $\mathbf{B}^{*(t)}$ or $\mathbf{C}^{*(t)}$ element; see (9). In practice, we can use all non-zero columns of $\mathbf{Z}_1^{(t)\top}$, however for our probabilistic analysis, we require an independent set of samples. We form one such set by selecting the first column from the first block, second column from the second block and so on; see Fig. 3. This results in a $L = \min(J, K)$ independent samples set for a given $\mathbf{Z}_1^{(t)\top}$. With this, and our assumptions on sparse factors ensure that the $L$ independent columns of $\mathbf{X}^{*(t)}$ ($\mathbf{X}_{\text{indep}}^{*(t)}$) belong to the distribution class $\mathcal{D}$ defined as follows.

**Definition 4 (Distribution class $\mathcal{D}$)** *The coefficient vector $\mathbf{x}^*$ belongs to an unknown distribution $\mathcal{D}$, where the support $S = \text{supp}(\mathbf{x}^*)$ is at most of size $s$, $\mathbf{Pr}[i \in S] = \Theta(s/m)$ and $\mathbf{Pr}[i, j \in S] = \Theta(s^2/m^2)$. Moreover, the distribution is normalized such that $\mathbf{E}[\mathbf{x}_i^* | i \in S] = 0$ and $\mathbf{E}[\mathbf{x}_i^{*2} | i \in S] = 1$, and when $i \in S$, $|\mathbf{x}_i^*| \geq C$ for some constant $C \leq 1$. In addition, the non-zero entries are sub-Gaussian and pairwise independent conditioned on the support.*

Further, the $(\epsilon_0, 2)$-closeness (Def. 1) ensures that the signed-support (defined below) of the coefficients are recovered correctly (with high probability).

**Definition 5** *The signed-support of a vector $\mathbf{x}$ is defined as $\text{sign}(\mathbf{x}) \cdot \text{supp}(\mathbf{x})$.*

**Scaling and Permutation Indeterminacy**: The unit-norm constraint on $\mathbf{A}^*$ implies that the scaling (including the sign) ambiguity only exists in the recovery of $\mathbf{B}^{*(t)}$ and $\mathbf{C}^{*(t)}$. To this end, we will regard our algorithm to be successful in the following sense.

**Definition 6 (Equivalence)** *Factorizations $[\![\mathbf{A}, \mathbf{B}, \mathbf{C}]\!]$ are considered equivalent up to scaling, i.e, $[\![\mathbf{A}, \mathbf{B}, \mathbf{C}]\!] = [\![\mathbf{A}^*, \mathbf{B}^* \mathbf{D}_{\sigma_b}, \mathbf{C}^* \mathbf{D}_{\sigma_c}]\!]$ where $\sigma_b(\sigma_c)$ is a vector of scalings (including signs) corresponding to columns of the factors $\mathbf{B}$ and $\mathbf{C}$, respectively.*

**Dictionary Factor Update Strategy**: We use an *approximate* (we use an estimate of $\mathbf{X}^{*(t)}$) gradient descent-based strategy (7) to update $\mathbf{A}^{(t)}$ by finding a direction $\mathbf{g}_i^{(t)}$ to ensure descent. Here, the $(\Omega(s/m), \Omega(m/s), 0)$-correlatedness (defined below) of the expected gradient vector is sufficient to make progress ("0" indicates no bias); see [19, 49–51].

**Definition 7** *A vector $\mathbf{g}_i^{(t)}$ is $(\rho_-, \rho_+, \zeta_t)$-correlated with a vector $\mathbf{z}^*$ if for any vector $\mathbf{z}^{(t)}$*

$$\langle \mathbf{g}_i^{(t)}, \mathbf{z}^{(t)} - \mathbf{z}^* \rangle \geq \rho_- \|\mathbf{z}^{(t)} - \mathbf{z}^*\|^2 + \rho_+ \|\mathbf{g}_i^{(t)}\|^2 - \zeta_t.$$

Our model assumptions can be formalized as follows, with which we state our main result.

**A.1** $\mathbf{A}^*$ is $\mu$-incoherent (Def. 2), where $\mu = \mathcal{O}(\log(n))$, $\|\mathbf{A}^*\| = \mathcal{O}(\sqrt{m/n})$ and $m = \mathcal{O}(n)$;

**A.2** $\mathbf{A}^{(0)}$ is $(\epsilon_0, 2)$-close to $\mathbf{A}^*$ as per Def. 1, and $\epsilon_0 = \mathcal{O}^*(1/\log(n))$;

**A.3** Factors $\mathbf{B}^{*(t)}$ and $\mathbf{C}^{*(t)}$ are respectively drawn from distributions $\Gamma_{\alpha, C}^{\text{sG}}$ and $\Gamma_{\beta}^{\text{Rad}}$ (Def.3);

**A.4** Sparsity controlling parameters $\alpha$ and $\beta$ obey $\alpha\beta = \mathcal{O}(\sqrt{n}/m\mu \log(n))$ for $m = \Omega(\log(\min(J, K))/\alpha\beta)$, resulting column sparsity $s$ of $\mathbf{S}^{*(t)}$ is $s = \mathcal{O}(\alpha\beta m)$;

**A.5** The dictionary update step-size satisfies $\eta_A = \Theta(m/s)$;

**A.6** The coefficient update step-size and threshold satisfy $\eta_x^{(r)} < c_1(\epsilon_t, \mu, n, s) = \widetilde{\Omega}(s/\sqrt{n}) < 1$ and $\tau^{(r)} = c_2(\epsilon_t, \mu, s, n) = \widetilde{\Omega}(s^2/n)$ for small constants $c_1$ and $c_2$.

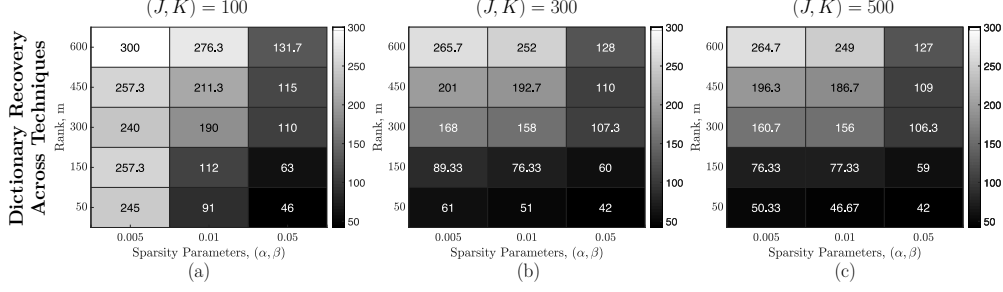

**Figure 4:** Number of iterations for convergence as a surrogate for data samples requirement [3]. Panels (a), (b), and (c) show the iterations taken by `TensorNOODL` to achieve a tolerance of $10^{-10}$ for $\mathbf{A}$ for $J=K=\{100, 300, 500\}$, respectively across ranks $m=\{50, 150, 300, 450, 600\}$ and $\alpha=\beta=\{0.005, 0.01, 0.05\}$, averaged across 3 Monte Carlo runs.

**Theorem 1 (Main Result)** *Suppose a tensor $\underline{\mathbf{Z}}^{(t)} \in \mathbb{R}^{n \times J \times K}$ provided to Alg. 1 at each iteration t admits a decomposition of the form* (1) *with factors $\mathbf{A}^* \in \mathbb{R}^{n \times m}$, $\mathbf{B}^{*(t)} \in \mathbb{R}^{J \times m}$ and $\mathbf{C}^{*(t)} \in \mathbb{R}^{K \times m}$ and $\min(J, K) = \Omega(ms^2)$. Further, suppose that the assumptions A.1-A.6 hold. Then, given $R = \Omega(\log(n))$, with probability at least $(1 - \delta_{alg})$ for some small constant $\delta_{alg}$, the estimate $\widehat{\mathbf{X}}^{(t)}$ at t-th iteration has the correct signed-support and satisfies*

$$(\widehat{\mathbf{X}}_{i,j}^{(t)} - \mathbf{X}_{i,j}^{*(t)})^2 \leq \zeta^2 := \mathcal{O}(s(1-\omega)^{t/2}\|\mathbf{A}_i^{(0)} - \mathbf{A}_i^*\|), \forall (i,j) \in \mathrm{supp}(\mathbf{X}^{*(t)}).$$

*Furthermore, for some $0 < \omega < 1/2$, the estimate $\mathbf{A}^{(t)}$ at t-th iteration satisfies*

$$\|\mathbf{A}_i^{(t)} - \mathbf{A}_i^*\|^2 \leq (1-\omega)^t \|\mathbf{A}_i^{(0)} - \mathbf{A}_i^*\|^2, \ \forall \ t = 1, 2, \dots$$

*Consequently, Alg. 2 recovers the supports of the sparse factors $\mathbf{B}^{*(t)}$ and $\mathbf{C}^{*(t)}$ correctly, and $\|\widehat{\mathbf{B}}_i^{(t)} - \mathbf{B}_i^{*(t)}\|_2 \leq \epsilon_B$ and $\|\widehat{\mathbf{C}}_i^{(t)} - \mathbf{C}_i^{*(t)}\|_2 \leq \epsilon_C$, where $\epsilon_B = \epsilon_C = \mathcal{O}(\frac{\zeta^2}{\alpha\beta})$.*

**Discussion**: Theorem 1 states the sufficient conditions under which, for an appropriate dictionary factor initialization (**A.2**), if the incoherent factor $\mathbf{A}^*$ columns are sufficiently spread out ensuring identifiability (**A.1**), the sparse factors $\mathbf{B}^{*(t)}$ and $\mathbf{C}^{*(t)}$ are appropriately sparse (**A.3** and **A.4**), and for appropriately chosen learning parameters (step sizes and threshold **A.5**∼**A.6**), then Alg. 1 succeeds whp. Such initializations can be achieved by existing algorithms and can also be used for model selection, i.e., determining $m$ i.e. *revealing rank*; see [19]. Also, from **A.4**, we observe that the sparsity $s$ (number of non-zeros) in a column of $\mathbf{S}^{*(t)}$) are critical for the success of the algorithm. Specifically, the upper-bound on $s$ keeps $s$ small for the success of dictionary learning, while the lower-bound on $m$ for given sparsity controlling probabilities($\alpha, \beta$) ensures that there are enough *independent* non-zero columns in $\mathbf{S}^{*(t)}$) for learning. In other words, this condition ensures that sparsity is neither too low (to avoid degeneracy) nor too high (for dictionary learning), requiring that the independent samples $L = \min(J, K) = \Omega(ms^2)$, wherein $s = \mathcal{O}(\alpha\beta m)$ whp.

## 5 Numerical Simulations

We evaluate `TensorNOODL` on synthetic and real-world data; more results in Appendix E.

### 5.1 Synthetic data evaluation

**Experimental set-up**: We compare `TensorNOODL` with online dictionary learning algorithms presented in [19] (`Arora(b)` (incurs bias) and `Arora(u)` (claim no bias)), and [20], which can be viewed as a variant of ALS (matricized) [4]. We analyze the recovery performance of the algorithms across different choices of tensor dimensions $J = K = \{100, 300, 500\}$ for a fixed $n = 300$, rank $m = \{50, 150, 300, 450, 600\}$, and the sparsity parameters $\alpha = \beta = \{0.005, 0.01, 0.05\}$ of factors $\mathbf{B}^{*(t)}$ and $\mathbf{C}^{*(t)}$, across 3 Monte-Carlo runs [5].

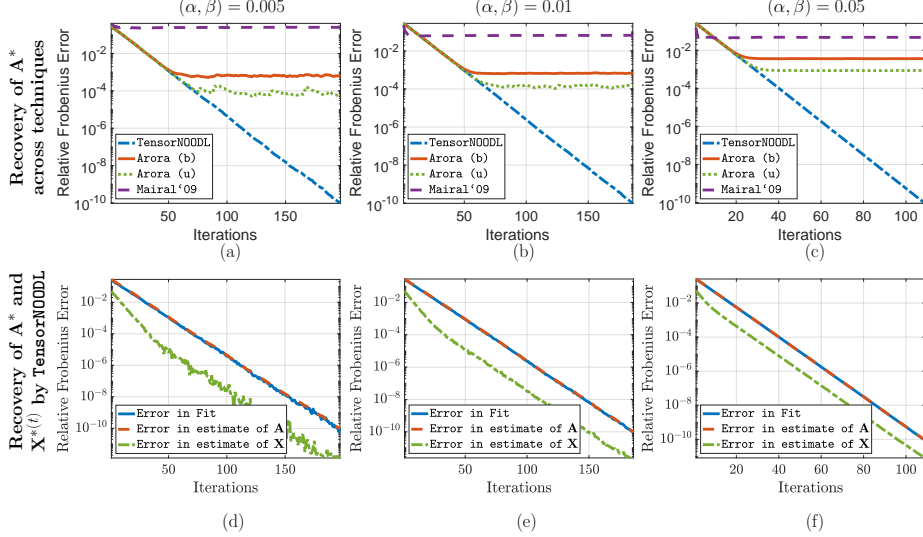

**Figure 5:** Linear convergence of `TensorNOODL`. Panels (a), (b), and (c) show the convergence properties of `TensorNOODL`, `Arora (b)`, `Arora (u)` and `Mairal'09` for the incoherent factor $\mathbf{A}$ recovery for $(\alpha, \beta) = 0.005, 0.01$ and $0.05$ respectively for $m = 450$, $(J, K) = 500$ and seed= 26. Panels (c), (d), and (e), show the recovery of $\mathbf{X}^{*(t)}$ (i.e. $\mathbf{B}^{*(t)}$ and $\mathbf{C}^{*(t)}$) $\mathbf{A}^{*}$, and the data fit (i.e., $\|\mathbf{Y}^{(t)} - \mathbf{A}^{(t)} \widehat{\mathbf{X}}^{(t)}\|_{\mathrm{F}} / \|\mathbf{Y}^{(t)}\|_{\mathrm{F}}$) for `TensorNOODL` corresponding to (a), (b), and (c), respectively.

We draw entries of $\mathbf{A}^{*} \in \mathbb{R}^{n \times m}$ from $\mathcal{N}(0, 1)$, and normalize its columns to be unit-norm. To form $\mathbf{A}^{(0)}$, we perturb $\mathbf{A}^{*}$ with random Gaussian noise and normalized its columns, such that it is column-wise $2/\log(n)$ away from $\mathbf{A}^{*}$ (**A.2**). To form $\mathbf{B}^{*(t)}$ (and $\mathbf{C}^{*(t)}$), we independently pick the non-zero locations with probability $\alpha$ (and $\beta$), and draw the values on the support from the Rademacher distribution[6]; see Appendix E.1 for details.

**Discussion**: We focus on the recovery of $\mathbf{X}^{*(t)}$ (including support recovery) since the performance of Alg. 2 solely depends on exact recovery of $\mathbf{X}^{*(t)}$. In Fig. 4, we analyze the samples requirement across different choices of the dimension $(J, K)$, rank $(m)$ and sparsity parameters $(\alpha, \beta)$ averaged across Monte Carlo runs using the total iterations $T$[4]. In line with theory, we observe a) in each panel the total iterations (to achieve tolerance $\epsilon_T$) decreases with increasing $(\alpha, \beta)$, and b) for a fixed rank and sparsity parameters the $T$ decreases with increasing $(J, K)$, these are both due to the increase in available data samples; also sample requirement increases with rank $m$. Furthermore, only `TensorNOODL` recovers the correct support of $\mathbf{X}^{*(t)}$, crucial for sparse factor recovery. Corroborating our theoretical results, `TensorNOODL` achieves orders of magnitude superior recovery at linear rate (Fig. 5) as compared to competing techniques both for the recovery of $\mathbf{A}^{*}$, and $\mathbf{X}^{*(t)}$. Moreover, since $\mathbf{X}^{*}$ columns can be estimated independently, `TensorNOODL` is scalable and can be implemented in highly distributed settings.

## 5.2 Real-world data evaluation

We consider a real data application in sports analytics. Additional real-data experiments for an email activity-based organizational behavior application are presented in Appendix E.2.1.

**NBA Shot Pattern Dataset**

We analyze weekly shot patterns of the 100 high scoring players ($80^{\text{th}}$ percentile) against 30 teams in the $2018 - 19$ regular season (27 weeks) of the National Basketball Association (NBA) league. The task is to identify specific shot patterns attempted by players against teams and cluster them from the weekly $100 \times 30 \times 120$ shot pattern tensor.

**Methodology**: We divide half-court into $10 \times 12$ blocks and sum-up all shots attempted by a player in a game from a particular block, and vectorize to form a shot pattern vector ($\mathbb{R}^{120}$) of a player against a particular opponent team. We use $2017 - 18$'s regular season

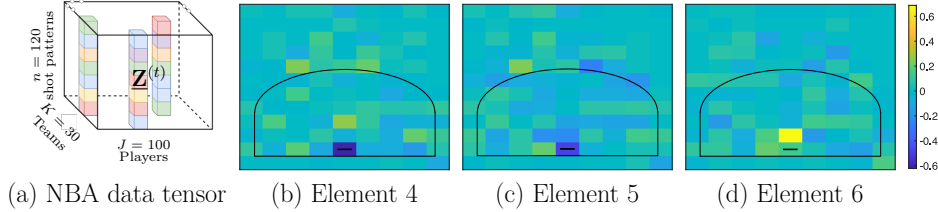

|  | (a) NBA data tensor | (b) Element 4 | (c) Element 5 | (d) Element 6 |

| Corresponding Sparse factor (Players) Coefficients | | | |
|---|---|---|---|
| **Player** | **Element 4** | **Element 5** | **Element 6** |
| James Harden | 0.1992 | 0.0678 | 0.2834 |
| Devin Booker | 0.0114 | 0.0104 | 0.4668 |

**Figure 6:** NBA Regular Season Shot Pattern data analysis. `TensorNOODL` clusters the players and the teams. Panel (a) show the structured tensor of interest $\mathbf{Z}^{(t)} \in \mathbb{R}^{n \times J \times K}$ for this application. There are 27 such tensors arriving every week of the season. Panels (b)-(d) show the three recovered dictionary factor elements shared by James Harden and Devin Booker (believed to have similar styles) during week 10 of the regular season $(2018 - 19)$.

data to initialize incoherent factor using [19], recovering 7 elements. The resulting tensor available at each week is shown in Fig. 6 (a).

**Discussion**: In Fig. 6 panels (b)-(d) we show the three recovered shot patterns and the corresponding weights for week-10. `TensorNOODL` reveals the similarity in shot selection of James Harden and Devin Booker, in line with the sports reports at the time [52, 53]. The shared elements show players' shot preference above the 3-point line (Fig. 6(a-b)) and at the rim (Fig. 6(c)); See Appendix E.2.2 for detailed results, and Appendix E.2.1 for evaluations on Enron data.

## 6 Discussion

**Summary**: Leveraging a matrix view of the tensor factorization task, we propose `TensorNOODL`, to the best of our knowledge, the first provable algorithm to achieve exact (up to scaling and permutations) *online* structured 3-way tensor factorization at a linear rate. Our analysis to untangle the Kronecker product dependence structure (induced by the matricized view) can be leveraged by other tensor factorization tasks.

**Limitations and Future Work**: We use probabilistic model assumptions which requires us to carefully identify independent samples. Although not an issue in practice, this leads to somewhat conservative results. Future work includes improving this sample efficiency.

**Conclusions**: We analyze an exciting modality where the tensor decomposition task can be reduced to that of matrix factorization. Such correspondences offer a way to establish strong convergence and recovery guarantees for structured tensor factorization tasks.

## Acknowledgments

The authors graciously acknowledge the support from the DARPA YFA, Grant N66001-14-1-4047. The authors would also like to express their gratitude to Prof. Nikos Sidiropoulos and Di Xiao for their helpful discussions. The research work was undertaken when Sirisha Rambhatla was a doctoral student at the Department of Electrical and Computer Engineering, University of Minnesota – Twin Cities, Minneapolis, MN.

## Broader Impact

This work explores the theoretical foundations behind the success of popular alternating minimization-based techniques for tensor factorization. Specifically, we propose an algorithm for accurate model recovery for a tensor factorization task which has applications in clustering and pattern recovery. Since clustering-based algorithms are used for identification of users for targeted advertising campaigns on social network platforms, potential use cases may target users based on their activity patterns. Nevertheless, understanding the theoretical aspects of machine learning algorithms is crucial for ensuring safety and trustworthiness in critical applications, and can in fact be used to mitigate effects of the very biases that these algorithms are prone to exacerbate.

## Footnotes

[2]The non-zero entries of $\mathbf{C}^{*(t)}$ can also be assumed to be drawn from a sub-Gaussian distribution (like $\mathbf{B}^{*(t)}$) at the expense of sparsity, incoherence, dimension(s), and sample complexity. Specifically when non-zero entries of $\mathbf{B}^{*(t)}$ and $\mathbf{C}^{*(t)}$ are drawn from sub-Gaussian distribution (as per $\Gamma_{\gamma,C}^{\mathrm{sG}}$), we will need the dictionary learning algorithm to work with the coefficient matrix $\mathbf{X}^{*(t)}$ (formed by product of entries of $\mathbf{B}^{*(t)}$ and $\mathbf{C}^{*(t)}$) which now has sub-Exponential non-zero entries.

[4] As discussed, the provable tensor factorization algorithms shown in Table 1, are suitable only for cases wherein all the factors obey same structural assumptions, and also are not online.

[4] `TensorNOODL` takes a fresh tensor at each $t$, we use $T$ as a surrogate for sample requirement.

[5] We fix $(J, K)$ & $(\alpha, \beta)$, but `TensorNOODL` can also be used with iteration-dependent parameters.

[6]Corresponding code is available at `https://github.com/srambhatla/TensorNOODL`.

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
