[Supplementary Material]

# Supplementary Material for Provable Online CP/PARAFAC Decomposition of a Structured Tensor via Dictionary Learning

**Sirisha Rambhatla** [1]
sirishar@usc.edu

**Xingguo Li** [2]
xingguol@cs.princeton.edu

**Jarvis Haupt** [3]
jdhaupt@umn.edu

[1] Computer Science Department, University of Southern California
[2] Computer Science Department, Princeton University
[3] Department of Electrical and Computer Engineering, University of Minnesota – Twin Cities

## Navigating Supplementary Material

We summarize the notation used in our work in Appendix A, including a list of frequently used symbols and their corresponding definitions. Next, in Appendix B, we present the proof of our main result, and organize the proofs of intermediate results in Appendix C; additional results used are listed in Appendix D for completeness. Furthermore, we show the detailed synthetic and real-world experimental results, along with how to reproduce them, in Appendix E. Corresponding code with specific recommendation on the parameter setting is available at https://github.com/srambhatla/TensorNOODL.

## A  Summary of Notation

In addition to the notation described in the manuscript, we use $\|\mathbf{M}\|$ and $\|\mathbf{M}\|_{\mathrm{F}}$ for the spectral and Frobenius norm, respectively, and $\|\mathbf{v}\|$, $\|\mathbf{v}\|_0$, and $\|\mathbf{v}\|_1$ to denote the $\ell_2$, $\ell_0$ (number of non-zero entries), and $\ell_1$ norm, respectively. In addition, we use $\mathbf{D}_{(\mathbf{v})}$ as a diagonal matrix with elements of a vector $\mathbf{v}$ on the diagonal. Given a matrix $\mathbf{M}$, we use $\mathbf{M}_{-i}$ to denote a resulting matrix without $i$-th column. Also note that, since we show that $\|\mathbf{A}_i^{(t)} - \mathbf{A}_i^*\| \leq \epsilon_t$ contracts in every step, therefore we fix $\epsilon_t, \epsilon_0 = \mathcal{O}^*(1/\log(n))$ in our analysis. We summarize the definitions of some frequently used symbols in our analysis in Table A.1 and A.2.

**Table A.1:** Frequently used symbols: Probabilities

**Probabilities**

| Symbol | Definition | Symbol | Definition |
|---|---|---|---|
| $\gamma$ | $\gamma := \alpha\beta$, where $\alpha(\beta)$ is the probability that an element $\mathbf{B}_{ij}^{*(t)}$ ($\mathbf{C}_{ij}^{*(t)}$) of $\mathbf{B}^{*(t)}$ ($\mathbf{C}^{*(t)}$) is non-zero. | $\delta_{\mathbf{B}_i}^{(t)}$ | $\delta_{\mathbf{B}_i}^{(t)} = \exp(-\frac{\epsilon^2 J\alpha}{2(1+\epsilon/3)})$ for any $\epsilon > 0$. |
| $\delta_{\mathcal{T}}^{(t)}$ | $\delta_{\mathcal{T}}^{(t)} = 2m \exp(-\frac{C^2}{\mathcal{O}^*(\epsilon_t^2)})$ | $\delta_{\beta}^{(t)}$ | $2s \exp(-\frac{1}{\mathcal{O}(\epsilon_t)})$ |
| $\delta_s^{(t)}$ | $\delta_s^{(t)} = \min(J,K) \exp(-\epsilon^2\alpha\beta m/2(1+\epsilon/3))$ for any $\epsilon > 0$ | $\delta_p^{(t)}$ | $\delta_p^{(t)} = \exp(-\frac{\epsilon^2}{2}L(1-(1-\gamma)^m))$ |
| $\delta_{\mathrm{IHT}}^{(t)}$ | $\delta_{\mathrm{IHT}}^{(t)} = \delta_{\mathcal{T}}^{(t)} + \delta_{\beta}^{(t)}$ | $\delta_{\mathrm{NOODL}}^{(t)}$ | $\delta_{\mathrm{NOODL}}^{(t)} = \delta_{\mathcal{T}}^{(t)} + \delta_{\beta}^{(t)} + \delta_{\mathrm{HW}} + \delta_{\mathbf{g}_i}^{(t)} + \delta_{\mathbf{g}}^{(t)}$ |
| $q_i$ | $q_i = \mathbf{Pr}[i \in S] = \Theta(\frac{s}{m})$ | $q_{i,j}$ | $q_{i,j} = \mathbf{Pr}[i,j \in S] = \Theta(\frac{s^2}{m^2})$ |
| $p_i$ | $p_i = \mathbf{E}[\mathbf{X}_{ij}^* \mathrm{sign}(\mathbf{X}_{ij})|\mathbf{X}_{ij}^* \neq 0]$ | $\delta_{\mathrm{HW}}^{(t)}$ | $\delta_{\mathrm{HW}}^{(t)} = \exp(-1/\mathcal{O}(\epsilon_t))$ |
| $\delta_{\mathbf{g}_i}^{(t)}$ | $\delta_{\mathbf{g}_i}^{(t)} = \exp(-\Omega(s))$ | $\delta_{\mathbf{g}}^{(t)}$ | $\delta_{\mathbf{g}}^{(t)} = (n+m)\exp(-\Omega(m\sqrt{\log(n)}))$ |

**Table A.2:** Frequently used symbols: Notation and Parameters

| Symbol | Definition | Symbol | Definition |
|---|---|---|---|
| $(\cdot)^*$ | Used to represent the ground-truth matrices. | $(\cdot)^{(t)}$, $\widehat{(\cdot)}^{(t)}$, and $\widehat{(\cdot)}$ | Used to represent the estimates formed by the algorithm. |
| $(\cdot)^{(t)}$ | The subscript "$t$" is used to represent the estimates at $t$-iteration of the online algorithm. | $\mathbf{X}^{(r)(t)}$ | The $r$-th IHT iterate at $t$-th iterate of the online algorithm. |
| $(\cdot)^{(r)}$ | The subscript "$r$" is used to represent the $r$-th IHT iterate. | $\widehat{\mathbf{X}}^{(t)}$ | The final IHT estimate at $(r = R)$, i.e., $\mathbf{X}^{(R)(t)}$ at the $t$-th iterate of the online algorithm. |
| $\mathbf{A}_i^{(t)}$ | $i$-th column of $\mathbf{A}^{(}t)$ (estimate of $\mathbf{A}^*$ at the $t$-th iteration of the online algorithm). | $\widehat{\mathbf{B}}^{(t)}$ $(\widehat{\mathbf{C}}^{(t)})$ | Estimate of $\mathbf{B}^{*(t)}$ $(\mathbf{C}^{*(t)})$ at the $t$-th iteration of the online algorithm. |
| $\mathbf{S}^{*(t)}$ | Transposed Khatri-Rao structured (sparse) matrix, $\mathbf{S}^{*(t)} = (\mathbf{C}^{*(t)} \odot \mathbf{B}^{*(t)})^\top$, its $i$-th row is given by $\mathbf{C}_i^{*(t)} \otimes \mathbf{B}_i^{*(t)}$. | $\mathbf{X}^{*(t)}$ | Sparse matrix formed by collecting non-zero columns of $\mathbf{S}^{*(t)}$. |
| $p$ | Number of columns in $\mathbf{X}^{*(t)}$, also the number of non-zero columns in $\mathbf{S}^{*(t)}$. | $\mathbf{Z}_1^{(t)\top}$ | Mode-1 unfolding of $\underline{\mathbf{Z}}^{(t)}$, $\mathbf{Z}_1^{(t)\top} = \mathbf{A}^*(\mathbf{C}^{*(t)} \odot \mathbf{B}^{*(t)})^\top$ at the $t$-th iteration of the online algorithm. |
| $\epsilon_t$ | Upper-bound on column-wise error at the $t$-th iterate,$\|\mathbf{A}_i^{(t)} - \mathbf{A}_i^*\| \leq \epsilon_t = \mathcal{O}^*(\frac{1}{\log(n)})$. | $\mu$ | The incoherence between the columns of the factor $\mathbf{A}^*$; see Def. 2. |
| $\mu_t$ | Incoherence between the columns of $\mathbf{A}^{(t)}$, $\frac{\mu_t}{\sqrt{n}} = \frac{\mu}{\sqrt{n}} + 2\epsilon_t$. | $\xi$ | The element-wise upper bound on the error between $\widehat{\mathbf{S}}_{ij}^{(t)}$ and $\mathbf{S}_{ij}^{*(t)}$, i.e., $|\mathbf{S}_{ij}^{*(t)} - \widehat{\mathbf{S}}_{ij}^{(t)}| \leq \xi$. |
| $s$ | The number of non-zeros in a column of $\mathbf{S}^{*(t)}$, also referred to as the *sparsity*. | $\alpha(\beta)$ | The probability that an element $\mathbf{B}_{ij}^{*(t)}$ ($\mathbf{C}_{ij}^{*(t)}$) of $\mathbf{B}^{*(t)}$ ($\mathbf{C}^{*(t)}$) is non-zero. |
| $\epsilon_B$ | Upper-bound on column-wise $\ell_2$-error in the estimate $\widehat{\mathbf{B}}^{(t)}$ at $t$-th iteration, i.e., $\|\widehat{\mathbf{B}}_i^{(t)} - \mathbf{B}_i^{*(t)}\| \leq \epsilon_B = \mathcal{O}(\frac{\xi^2}{\alpha\beta})$. | $\epsilon_C$ | Upper-bound on column-wise $\ell_2$-error in the estimate $\widehat{\mathbf{C}}^{(t)}$ at $t$-th iteration, i.e., $\|\widehat{\mathbf{C}}_i^{(t)} - \mathbf{C}_i^{*(t)}\| \leq \epsilon_C = \mathcal{O}(\frac{\xi^2}{\alpha\beta})$. |
| $R$ | The total number of IHT steps at the $t$-th iteration of the online algorithm. | $T$ | Total number of online iterations. |
| $\delta_R$ | Decay parameter for final IHT step at every $t$, $\text{ceil}(\frac{\log(\frac{1}{\delta_R})}{\log(1-\eta_x)}) \leq R$, where $\eta_x$ is the step-size parameter for the IHT step. | $\delta_T$ | Element-wise target error tolerance for final estimate (at $t = T$) of $\mathbf{X}^{*(T)}$, $|\widehat{\mathbf{X}}_{ij}^{(T)} - \mathbf{X}_{ij}^{*(T)}| \leq \delta_T \forall i \in \text{supp}(\mathbf{X}^{*(T)})$. |
| $C$ | Lower-bound on $\mathbf{X}_{ij}^*$, $|\mathbf{X}_{ij}^{*(t)}| \geq C$ for $(i,j) \in \text{supp}(\mathbf{X}^{*(t)})$ and $C \leq 1$ | $L$ | $L := \min(J, K)$ |

# B  Proof of Theorem 1

In this section, we present the details of the analysis pertaining to our main result.

**Theorem 1 [Main Result]** *Suppose a tensor $\underline{\mathbf{Z}}^{(t)} \in \mathbb{R}^{n \times J \times K}$ provided to Algorithm 1 at each iteration $t$ admits a decomposition of the form (1) with factors $\mathbf{A}^* \in \mathbb{R}^{n \times m}$, $\mathbf{B}^{*(t)} \in \mathbb{R}^{J \times m}$ and $\mathbf{C}^{*(t)} \in \mathbb{R}^{K \times m}$ and $\min(J, K) = \Omega(ms^2)$. Further, suppose that the assumptions A.1-A.6 hold. Then, given $R = \Omega(\log(n))$, with probability at least $(1 - \delta_{alg})$ for some small constant $\delta_{alg}$, the coefficient estimate $\widehat{\mathbf{X}}^{(t)}$ at $t$-th iteration has the correct signed-support and satisfies*

$$(\widehat{\mathbf{X}}_{i,j}^{(0)} - \mathbf{X}_{i,j}^{*(t)})^2 \leq \zeta^2 := \mathcal{O}(s(1-\omega)^{t/2}\|\mathbf{A}_i^{(0)} - \mathbf{A}_i^*\|), \ \textit{for all } (i,j) \in \text{supp}(\mathbf{X}^{*(t)}).$$

*Furthermore, for some $0 < \omega < 1/2$, the estimate $\mathbf{A}^{(t)}$ at $t$-th iteration satisfies*

$$\|\mathbf{A}_i^{(t)} - \mathbf{A}_i^*\|^2 \leq (1-\omega)^t\|\mathbf{A}_i^{(0)} - \mathbf{A}_i^*\|^2, \ \textit{for all } t = 1, 2, \ldots.$$

*Consequently, Algorithm 2 recovers the supports of the sparse factors $\mathbf{B}^{*(t)}$ and $\mathbf{C}^{*(t)}$ correctly, and $\|\widehat{\mathbf{B}}_i^{(t)} - \mathbf{B}_i^{*(t)}\|_2 \leq \epsilon_B$ and $\|\widehat{\mathbf{C}}_i^{(t)} - \mathbf{C}_i^{*(t)}\|_2 \leq \epsilon_C$, where $\epsilon_B = \epsilon_C = \mathcal{O}(\frac{\zeta^2}{\alpha\beta})$.*

*Here, $\delta_{alg} = \delta_s + \delta_p^{(t)} + \delta_{\mathbf{B}_i}^{(t)} + \delta_{\text{NOODL}}$. Further, $\delta_{NOODL}^{(t)} = \delta_{\mathcal{T}}^{(t)} + \delta_\beta^{(t)} + \delta_{\text{HW}} + \delta_{\mathbf{g}_i}^{(t)} + \delta_{\mathbf{g}}^{(t)}$, where $\delta_{\mathcal{T}}^{(t)} = 2m \exp(-C^2/\mathcal{O}^*(\epsilon_t^2))$, $\delta_\beta^{(t)} = 2s \exp(-1/\mathcal{O}(\epsilon_t))$, $\delta_{\text{HW}}^{(t)} = \exp(-1/\mathcal{O}(\epsilon_t))$, $\delta_{\mathbf{g}_i}^{(t)} = \exp(-\Omega(s))$, $\delta_{\mathbf{g}}^{(t)} = (n + m) \exp(-\Omega(m\sqrt{\log(n)}))$. Furthermore, $\delta_s^{(t)} = \min(J, K) \exp(-\epsilon^2 \alpha\beta m/2(1 + \epsilon/3))$ for any $\epsilon > 0$, $\delta_p^{(t)} = \exp(-\frac{\epsilon^2}{2}L(1 - (1 - \gamma)^m))$, and $\delta_{\mathbf{B}_i}^{(t)} = \exp(-\frac{\epsilon^2 J\alpha}{2(1+\epsilon/3)})$ for any $\epsilon > 0$. Also, $\|\mathbf{A}_i^{(t)} - \mathbf{A}_i^*\| \leq \epsilon_t$.*

### Proof of Theorem 1

The proof procedure relies on analyzing three main steps of Alg. 1 – 1) estimating the $\mathbf{X}^{*(t)}$ reliably corresponding to $\underline{\mathbf{Z}}^{(t)}$, 2) using $\mathbf{X}^{(t)}$ to estimate the factors $\mathbf{B}^*$ and $\mathbf{C}^*$, and 3) making progress on the estimate of $\mathbf{A}^*$ at every iteration $t$ of the online algorithm.

### Estimating the $\mathbf{X}^{*(t)}$ reliably

The sparse matrix $\mathbf{X}^{*(t)}$ is formed by collecting the non-zero columns of $\mathbf{S}^{*(t)} := (\mathbf{C}^{*(t)} \odot \mathbf{B}^{*(t)})^\top$ corresponding to $\underline{\mathbf{Z}}^{(t)}$. The sparsity pattern of $\mathbf{X}^{*(t)}$ columns encodes the sparsity patterns of columns of $\mathbf{B}^{*(t)}$ and $\mathbf{C}^{*(t)}$. As a result, recovering the support of $\mathbf{X}^{*(t)}$ *exactly* is crucial to recover $\mathbf{B}^*$ and $\mathbf{C}^*$. Furthermore, recovering the signed-support is also essential for making progress on the dictionary factor. We begin by characterizing the number of non-zeros ($s$) in a column of $\mathbf{S}^{*(t)}$ ($\mathbf{X}^{*(t)}$). The number of non-zeros in a column of $\mathbf{S}^{*(t)}$ are dependent on the non-zero elements of $\mathbf{B}^*$ and $\mathbf{C}^*$. Since each element of $\mathbf{B}^*$ ($\mathbf{C}^*$) is non-zero with probability $\alpha(\beta)$, the upper-bound on the sparsity ($s$) of $\mathbf{S}^{*(t)}$ column is given by the following lemma.

**Lemma 1** *If $m = \Omega(\log(\min(J, K))/\alpha\beta)$ then with probability at least $(1 - \delta_s^{(t)})$ the number of non-zeros $s$, in a column of $\mathbf{S}^{*(t)}$ are upper-bounded as $s = \mathcal{O}(\alpha\beta m)$, where $\delta_s^{(t)} = \min(J, K) \exp(-\epsilon^2 \alpha\beta m/2(1 + \epsilon/3))$ for any $\epsilon > 0$.*

In line with our intuition, the sparsity scales with the parameters $\alpha$, $\beta$ and $m$.

Next, we focus on the Iterative Hard Thresholding (IHT) phase of the algorithm; Similar results were established in [11, Lemma 1-4]. Here, the first step includes recovering the correct signed-support (Def. 5) of $\mathbf{X}^{*(t)}$ given an estimate $\mathbf{A}^{(0)}$, which is $(\epsilon_0, 2)$-near to $\mathbf{A}^*$ for $\epsilon_0 = \mathcal{O}^*(1/\log(n))$; see Def. 1. To this end, we leverage the following lemma, to guarantee that the initialization step correctly recovers the signed-support with probability at least $(1 - \delta_{\mathcal{T}}^{(t)})$, for $\delta_{\mathcal{T}}^{(t)} = 2m \exp(-\frac{C^2}{\mathcal{O}^*(\epsilon_t^2)})$.

**Lemma 2 (Signed-support recovery)** *Suppose $\mathbf{A}^{(t)}$ is $\epsilon_t$-close to $\mathbf{A}^*$. Then, if $\mu = \mathcal{O}(\log(n))$, $s = \mathcal{O}^*(\sqrt{n}/\mu\log(n))$, and $\epsilon_t = \mathcal{O}^*(1/\sqrt{\log(m)})$, with probability at least $(1 -$*

$\delta_{\mathcal{T}}^{(t)}$) *for each random sample* $\mathbf{y} = \mathbf{A}^* \mathbf{x}^*$:

$$\text{sign}(\mathcal{T}_{C/2}((\mathbf{A}^{(t)})^\top \mathbf{y}) = \text{sign}(\mathbf{x}^*),$$

*where* $\delta_{\mathcal{T}}^{(t)} = 2m \exp(-\frac{C^2}{\mathcal{O}^*(\epsilon_t^2)})$.

Using Lemma 1 and 2 we also arrive at the condition that $s = \mathcal{O}(\alpha \beta m) = \mathcal{O}^* \sqrt{n}/\mu \log(n)$, formalized as **A.4**. We now use the following result to ensure that each step of the IHT stage preserves the correct signed-support. Lemma 3, states the conditions on the step size parameter $\eta_x^{(r)}$, and the threshold $\tau^{(r)}$, such that the IHT-step preserves the correct signed-support with probability $\delta_{\text{IHT}}^{(t)}$, for $\delta_{\text{IHT}}^{(t)} = 2m \exp(-\frac{C^2}{\mathcal{O}^*(\epsilon_t^2)}) + 2s \exp(-\frac{1}{\mathcal{O}(\epsilon_t)})$.

**Lemma 3 (IHT update step preserves the correct signed-support**) *Suppose* $\mathbf{A}^{(t)}$ *is* $\epsilon_t$-*close to* $\mathbf{A}^*$, $\mu = \mathcal{O}(\log(n))$, $s = \mathcal{O}^*(\sqrt{n}/\mu \log(n))$, *and* $\epsilon_t = \mathcal{O}^*(1/\log(m))$ *Then, with probability at least* $(1 - \delta_\beta^{(t)} - \delta_{\mathcal{T}}^{(t)})$, *each iterate of the IHT-based coefficient update step shown in* (6) *has the correct signed-support, if for a constant* $c_1^{(r)}(\epsilon_t, \mu, s, n) = \widetilde{\Omega}(k^2/n)$, *the step size is chosen as* $\eta_x^{(r)} \leq c_1^{(r)}$, *and the threshold* $\tau^{(r)}$ *is chosen as*

$$\tau^{(r)} = \eta_x^{(r)}(t_\beta + \frac{\mu_t}{\sqrt{n}} \|\mathbf{x}^{(r-1)} - \mathbf{x}^*\|_1) := c_2^{(r)}(\epsilon_t, \mu, s, n) = \widetilde{\Omega}(s^2/n),$$

*for some constants* $c_1$ *and* $c_2$. *Here,* $t_\beta = \mathcal{O}(\sqrt{s \epsilon_t})$, $\delta_{\mathcal{T}}^{(t)} = 2m \exp(-\frac{C^2}{\mathcal{O}^*(\epsilon_t^2)})$ *,and* $\delta_\beta^{(t)} = 2s \exp(-\frac{1}{\mathcal{O}(\epsilon_t)})$.

Lemma 3 establishes condition on correct signed-support recovery by the IHT stage. We now leverage the following result, Lemma 4 to quantify the error incurred by $\widehat{\mathbf{X}}^{(t)}$ at the end of the $R$ IHT steps, i.e., $|\mathbf{X}_{ij}^{*(t)} - \widehat{\mathbf{X}}_{ij}^{(t)}| = |\mathbf{S}_{ij}^{*(t)} - \widehat{\mathbf{S}}_{ij}^{(t)}| \leq \xi$.

**Lemma 4 (Upper-bound on the error in coefficient estimation)** *With probability at least* $(1 - \delta_\beta^{(t)} - \delta_{\mathcal{T}}^{(t)})$ *the error incurred by each element* $(i_1, j_1) \in \text{supp}(\mathbf{X}^{*(t)})$ *of the coefficient estimate is upper-bounded as*

$$|\widehat{\mathbf{X}}_{i_1 j_1}^{(t)} - \mathbf{X}_{i_1 j_1}^{*(t)}| \leq \mathcal{O}(t_\beta) + \left( (R+1)s\eta_x \frac{\mu_t}{\sqrt{n}} \max_{(i,j)} |\mathbf{X}_{ij}^{(0)(t)} - \mathbf{X}_{ij}^{*(t)}| + |\mathbf{X}_{i_1 j_1}^{(0)(t)} - \mathbf{X}_{i_1 j_1}^{*(t)}| \right) \delta_R = \mathcal{O}(t_\beta)$$

*where* $t_\beta = \mathcal{O}(\sqrt{s \epsilon_t})$, $\delta_R := (1 - \eta_x + \eta_x \frac{\mu_t}{\sqrt{n}})^R$, $\delta_{\mathcal{T}}^{(t)} = 2m \exp(-\frac{C^2}{\mathcal{O}^*(\epsilon_t^2)})$, $\delta_\beta^{(t)} = 2s \exp(-\frac{1}{\mathcal{O}(\epsilon_t)})$, *and* $\mu_t$ *is the incoherence between the columns of* $\mathbf{A}^{(t)}$.

Also, the corresponding expression for $\widehat{\mathbf{X}}^{(t)}$, which facilitates the analysis of the dictionary updates, is given by Lemma 5.

**Lemma 5 (Expression for the coefficient estimate at the end of $R$-th IHT iteration)** *With probability at least* $(1 - \delta_{\mathcal{T}}^{(t)} - \delta_\beta^{(t)})$ *the $i$-th element of the coefficient estimate, for each* $i \in \text{supp}(\mathbf{x}^*)$, *is given by*

$$\widehat{\mathbf{x}}_i := \mathbf{x}_i^{(R)} = \mathbf{x}_i^*(1 - \lambda_i^{(t)}) + \vartheta_i^{(R)}.$$

*Here,* $|\vartheta_i^{(R)}| = \mathcal{O}(t_\beta)$, *where* $t_\beta = \mathcal{O}(\sqrt{s \epsilon_t})$. *Further,* $\lambda_i^{(t)} = |\langle \mathbf{A}_i^{(t)} - \mathbf{A}_i^*, \mathbf{A}_i^* \rangle| \leq \frac{\epsilon_t^2}{2}$, $\delta_{\mathcal{T}}^{(t)} = 2m \exp(-\frac{C^2}{\mathcal{O}^*(\epsilon_t^2)})$ *and* $\delta_\beta^{(t)} = 2s \exp(-\frac{1}{\mathcal{O}(\epsilon_t)})$.

Interestingly, Lemma 4 shows that the error in the non-zero elements of $\widehat{\mathbf{X}}$ only depends on the error in the incoherent factor (dictionary) $\mathbf{A}^{(t)}$, which results in the following expression for $\xi^2$.

$$\xi^2 := \mathcal{O}(s(1-\omega)^{t/2} \|\mathbf{A}_i^{(0)} - \mathbf{A}_i^*\|), \text{ for all } (i,j) \in \text{supp}(\mathbf{X}^*). \tag{1}$$

Therefore, if the column-wise error in the dictionary decreases at each iteration $t$, then the IHT-based sparse matrix estimates also improve progressively.

**Recover Sparse Factors B* and C* via Alg.2**

The results for the IHT-stage are foundational for the recovery of the sparse tensor factors $\mathbf{B}^{*(t)}$ and $\mathbf{C}^{*(t)}$ since they a) ensure correct signed-support recovery, guaranteed by Lemma 3 and b) establish an upper-bound on the estimation error in $\widehat{\mathbf{S}}^{(t)}$. With these results, we now establish the correctness of Alg. 2 given an entry-wise $\zeta$-close estimate of $\mathbf{S}^{*(t)}$, $|\widehat{\mathbf{S}}_{ij}^{(t)} - \mathbf{S}_{ij}^{*(t)}| \leq \zeta$ given by the IHT stage. This procedure recovers the sparse factors $\mathbf{B}^{*(t)}$ and $\mathbf{C}^{*(t)}$, given element-wise $\xi$-close estimate $\widehat{\mathbf{S}}$ of $\mathbf{S}^{*(t)}$. The following lemma establishes recovery guarantees on the sparse factors using the SVD-based Alg. 2, up to sign and scaling ambiguity.

**Lemma 6** *Suppose the input $\widehat{\mathbf{S}}^{(t)}$ to Alg. 2 is entry-wise $\zeta$ close to $\mathbf{S}^{*(t)}$, i.e., $|\widehat{\mathbf{S}}_{ij}^{(t)} - \mathbf{S}_{ij}^{*(t)}| \leq$ $\zeta$ and has the correct signed-support as $\mathbf{S}^{*(t)}$. Then with probability atleast $(1 - \delta_{\text{IHT}}^{(t)} - \delta_{\mathbf{B}_i}^{(t)})$, both $\widehat{\mathbf{B}}_i^{(t)}$ and $\widehat{\mathbf{C}}_i^{(t)}$ have the correct support, and $\left\| \frac{\mathbf{B}_i^{*(t)}}{\|\mathbf{B}_i^{*(t)}\|} - \pi_i \frac{\widehat{\mathbf{B}}_i^{(t)}}{\|\widehat{\mathbf{B}}_i^{(t)}\|} \right\| = \mathcal{O}(\zeta^2)$ and $\left\| \frac{\mathbf{C}_i^{*(t)*}}{\|\mathbf{C}_i^{*(t)}\|} - \pi_i \frac{\widehat{\mathbf{C}}_i^{(t)}}{\|\widehat{\mathbf{C}}_i^{(t)}\|} \right\| = \mathcal{O}(\zeta^2)$, where $\delta_{\text{IHT}}^{(t)} = 2m \exp(-\frac{C^2}{\mathcal{O}^*(\epsilon_t^2)}) + 2s \exp(-\frac{1}{\mathcal{O}(\epsilon_t)})$ for $\|\mathbf{A}_i^{(t)} - \mathbf{A}_i^*\| \leq \epsilon_t$, and $\delta_{\mathbf{B}_i}^{(t)} = \exp(-\frac{\epsilon^2 J\alpha}{2(1+\epsilon/3)})$ for any $\epsilon > 0$.*

Here, we have used $\delta_{\text{IHT}}^{(t)} = \delta_\beta^{(t)} + \delta_{\mathcal{T}}^{(t)}$ for simplicity.

**Update Dictionary Factor $\mathbf{A}^{(t)}$**

The update of the dictionary factor involves concentration results which rely on an independent set of data samples. For this, notice that the $i$-th row of $\mathbf{S}^{*(t)}$ can be written as $(\mathbf{C}_i^{*(t)} \otimes \mathbf{B}_i^{*(t)})^\top$. Now, since $\mathbf{B}^{*(t)}$ and $\mathbf{C}^{*(t)}$ are sparse, there are a number of columns in $\mathbf{S}^{*(t)}$ which are degenerate (all-zeros). As a result, the corresponding data samples (columns of $\mathbf{Z}_1^{(t)\top}$) are also degenerate, and cannot be used for learning. Furthermore, due to the dependence structure in $\mathbf{S}^{*(t)}$ (discussed in section 4) some of the data samples are dependent on each other, and at least from the theoretical perspective, are not eligible for the learning process. Therefore, we characterize the expected number of viable data samples in the following lemma.

**Lemma 7** *For $L = \min(J, K)$, $\gamma = \alpha\beta$, and any $\epsilon > 0$ and suppose we have*

$$L \geq \frac{2}{(1-(1-\gamma)^m)\epsilon^2} \log(\frac{1}{\delta_p^{(t)}}),$$

*then with probability at least $(1 - \delta_p)$,*

$$p = L(1 - (1-\gamma)^m),$$

*where $\delta_p^{(t)} = \exp(-\frac{\epsilon^2}{2}L(1-(1-\gamma)^m))$.*

Here, we observe that the number of viable samples increase with number of independent samples $L = \min(J, K)$, sparsity parameter $\gamma = \alpha\beta$, and rank of the decomposition $m$. To recover the incoherent (dictionary) factor $\mathbf{A}^*$, we follow analysis similar to [11, Lemma 5-9]. Here, we first develop an expression for the expected gradient vector in Lemma 8.

**Lemma 8 (Expression for the expected gradient vector)** *Suppose that $\mathbf{A}^{(t)}$ is $(\epsilon_t, 2)$-near to $\mathbf{A}^*$. Then, the dictionary update step in Alg. 1 amounts to the following for the $j$-th dictionary element*

$$\mathbf{E}[\mathbf{A}_j^{(t+1)}] = \mathbf{A}_j^{(t)} + \eta_A \mathbf{g}_j^{(t)},$$

*where for a small $\widetilde{\gamma}$, $\mathbf{g}_j^{(t)}$ is given by*

$$\mathbf{g}_j^{(t)} = q_j p_j \big( (1 - \lambda_j^{(t)})\mathbf{A}_j^{(t)} - \mathbf{A}_j^* + \frac{1}{q_j p_j}\Delta_j^{(t)} \pm \widetilde{\gamma} \big),$$

*$\lambda_j^{(t)} = |\langle \mathbf{A}_j^{(t)} - \mathbf{A}_j^*, \mathbf{A}_j^* \rangle|$, and $\Delta_j^{(t)} := \mathbf{E}[\mathbf{A}_S^{(t)} \vartheta_S^{(R)} \text{sign}(\mathbf{x}_j^*)]$, where $\|\Delta_j^{(t)}\| = \mathcal{O}(\sqrt{m} q_{i,j} p_j \epsilon_t \|\mathbf{A}^{(t)}\|)$.*

Since we use empirical gradient estimate, the following lemma establishes that the empirical gradient vector concentrates around its mean, and that it make progress at each step.

**Lemma 9 (Concentration of the empirical gradient vector)** *Given $p = \widetilde{\Omega}(mk^2)$ samples, the empirical gradient vector estimate corresponding to the $i$-th dictionary element, $\widehat{\mathbf{g}}_i^{(t)}$ concentrates around its expectation, i.e.,*

$$\|\widehat{\mathbf{g}}_i^{(t)} - \mathbf{g}_i^{(t)}\| \leq o(\tfrac{s}{m}\epsilon_t).$$

*with probability at least $(1 - \delta_{\mathbf{g}_i}^{(t)} - \delta_{\beta}^{(t)} - \delta_{\mathcal{T}}^{(t)} - \delta_{\mathrm{HW}}^{(t)})$, where $\delta_{\mathbf{g}_i}^{(t)} = \exp(-\Omega(s))$.*

We then leverage Lemma 10 to show that the empirical gradient vector $\widehat{\mathbf{g}}_j^{(t)}$ is correlated with the descent direction (see Def. 7), which ensures that the dictionary estimate makes progress at each iteration of the online algorithm.

**Lemma 10 (Empirical gradient vector is correlated with the descent direction)** *Suppose $\mathbf{A}^{(t)}$ is $(\epsilon_t, 2)$-near to $\mathbf{A}^*$, $s = \mathcal{O}(\sqrt{n})$ and $\eta_A = \mathcal{O}(m/s)$. Then, with probability at least $(1 - \delta_{\mathcal{T}}^{(t)} - \delta_{\beta}^{(t)} - \delta_{\mathrm{HW}}^{(t)} - \delta_{\mathbf{g}_i}^{(t)})$ the empirical gradient vector $\widehat{\mathbf{g}}_j^{(t)}$ is $(\Omega(k/m), \Omega(m/k), 0)$-correlated with $(\mathbf{A}_j^{(t)} - \mathbf{A}_j^*)$, and for any $t \in [T]$,*

$$\|\mathbf{A}_j^{(t+1)} - \mathbf{A}_j^*\|^2 \leq (1 - \rho_-\eta_A)\|\mathbf{A}_j^{(t)} - \mathbf{A}_j^*\|^2.$$

This step also requires closeness that the estimate $\mathbf{A}^{(t)}$ and $\mathbf{A}^*$ are close, both column-wise and in the spectral norm sense, as per Def 1. To this end, we show that the updated dictionary matrix maintain the closeness property. For this, we first show that the gradient matrix concentrates around its mean in Lemma 11.

**Lemma 11 (Concentration of the empirical gradient matrix)** *With probability at least $(1 - \delta_{\beta}^{(t)} - \delta_{\mathcal{T}}^{(t)} - \delta_{\mathrm{HW}}^{(t)} - \delta_{\mathbf{g}}^{(t)})$, $\|\widehat{\mathbf{g}}^{(t)} - \mathbf{g}^{(t)}\|$ is upper-bounded by $\mathcal{O}^*(\tfrac{s}{m}\|\mathbf{A}^*\|)$, where $\delta_{\mathbf{g}}^{(t)} = (n + m)\exp(-\Omega(m\sqrt{\log(n)})$.*

Further, the closeness property is maintained, as shown below.

**Lemma 12 ($\mathbf{A}^{(t+1)}$ maintains closeness)** *Suppose $\mathbf{A}^{(t)}$ is $(\epsilon_t, 2)$ near to $\mathbf{A}^*$ with $\epsilon_t = \mathcal{O}^*(1/\log(n))$, and number of samples used in step $t$ is $p = \widetilde{\Omega}(ms^2)$, then with probability at least $(1 - \delta_{\mathcal{T}}^{(t)} - \delta_{\beta}^{(t)} - \delta_{\mathrm{HW}}^{(t)} - \delta_{\mathbf{g}}^{(t)})$, $\mathbf{A}^{(t+1)}$ satisfies $\|\mathbf{A}^{(t+1)} - \mathbf{A}^*\| \leq 2\|\mathbf{A}^*\|$.*

Therefore, the recovery of factor $\mathbf{A}^*$, and the sparse-structured matrix $\mathbf{X}^*$ succeeds with probability $\delta_{\mathrm{NOODL}}^{(t)} = \delta_{\mathcal{T}}^{(t)} + \delta_{\beta}^{(t)} + \delta_{\mathrm{HW}} + \delta_{\mathbf{g}_i}^{(t)} + \delta_{\mathbf{g}}^{(t)}$, where $\delta_{\mathcal{T}}^{(t)} = 2m\exp(-C^2/\mathcal{O}^*(\epsilon_t^2))$, $\delta_{\beta}^{(t)} = 2s\exp(-1/\mathcal{O}(\epsilon_t))$, $\delta_{\mathrm{HW}}^{(t)} = \exp(-1/\mathcal{O}(\epsilon_t))$, $\delta_{\mathbf{g}_i}^{(t)} = \exp(-\Omega(s))$, $\delta_{\mathbf{g}}^{(t)} = (n + m)\exp(-\Omega(m\sqrt{\log(n)})$.

Further, from Lemma 1, we have that the columns of $\mathbf{S}^{*(t)}$ are $s = \mathcal{O}(\alpha\beta m)$ sparse with probability $(1 - \delta_s^{(t)})$, where $\delta_s^{(t)} = \min(J, K)\exp(-\epsilon^2\alpha\beta m/2(1+\epsilon/3))$ for any $\epsilon > 0$, and that with probability at least $(1 - \delta_p)$, the number of data samples $p = L(1 - (1 - \gamma)^m)$, where $\delta_p^{(t)} = \exp(-\tfrac{\epsilon^2}{2}L(1 - (1 - \gamma)^m))$ using Lemma 1. Furthermore, from Lemma 6, we know that Alg. 2 (which only relies on recovery of $\mathbf{X}^{*(t)}$) succeeds in recovering $\mathbf{B}^{*(t)}$ and $\mathbf{C}^{*(t)}$ (up to permutation and scaling) with probability $(1 - \delta_{\mathbf{B}_i}^{(t)})$, where $\delta_{\mathbf{B}_i}^{(t)} = \exp(-\tfrac{\epsilon^2 J\alpha}{2(1+\epsilon/3)})$ for any $\epsilon > 0$. Combining all these results we have that, Alg. 1 succeeds with probability $(1 - \delta_{alg})$, where $\delta_{alg} = \delta_s + \delta_p^{(t)} + \delta_{\mathbf{B}_i}^{(t)} + \delta_{\mathrm{NOODL}}$. Also, the total run time of the algorithm is $\mathcal{O}(mnp\log(1/\delta_R)\max(\log(1/\epsilon_T), \log(\sqrt{(s)}/\delta_T))$ for $p = \Omega(ms^2)$. Hence, our main result.

**A note on independent sample requirement:** Since the IHT-based coefficient operates independently on each column of $\mathbf{Y}^{(t)}$ (the non-zero columns of $\mathbf{Z}_1^{(t)\top}$), the dependence structure of $\mathbf{S}^{*(t)}$ does not affect this stage. For the dictionary update (in theory) we only use the independent columns of $\mathbf{Y}^{(t)}$, these can be inferred using $J$ and $K$, and

corresponding induced transposed Khatri-Rao structure. In practice however, we don't need to throw away any samples, this is purely to ensure that the independence assumption holds for our finite sample analysis of the algorithm. ∎

## C    Proof of Intermediate Results

**Lemma 1** *If $m = \Omega(\log(\min(J, K))/\alpha\beta)$ then with probability at least $(1 - \delta_s^{(t)})$ the number of non-zeros, $s$, in a column of $\mathbf{S}^{*(t)}$ are upper-bounded as $s = \mathcal{O}(\alpha\beta m)$, where $\delta_s^{(t)} = \min(J, K)\exp(-\epsilon^2\alpha\beta m/2(1 + \epsilon/3))$ for any $\epsilon > 0$.*

**Proof of Lemma 1** Consider a column of the transposed Khatri-Rao structured matrix $\mathbf{S}^{*(t)}$ defined as $\mathbf{S}^{*(t)} = (\mathbf{C}^{*(t)} \odot \mathbf{B}^{*(t)})^\top$. Here, since the entries of factors $\mathbf{B}^{*(t)}$ and $\mathbf{C}^{*(\mathbf{t})}$ are independently non-zero with probability $\alpha$ and $\beta$, respectively, each entry of a column of $\mathbf{S}^{*(t)}$ is independently non-zero with probability $\gamma = \alpha\beta$, i.e., $\mathbb{1}_{|\mathbf{S}_{ij}^{*(t)}|>0} \sim \text{Bernoulli}(\gamma)$.

As a result, the number of non-zero elements in a column of $\mathbf{S}^{*(t)}$ are Binomial$(m, \gamma)$.

Now, let $\mathbf{s}_{ij}$ be the indicator for the $(i, j)$ element of $\mathbf{S}^{*(t)}$ being non-zero, defined as

$$\mathbf{s}_{ij} = \mathbb{1}_{|\mathbf{S}_{ij}^{*(t)}|>0}.$$

Then, the expected number of non-zeros (sparsity) in the $j$-th column of $\mathbf{S}^{*(t)}$ are given by

$$\mathbf{E}[\sum_{i=1}^m \mathbf{s}_{ij}] = \gamma m.$$

Since, $\gamma$ can be small, we use Lemma 13(a) [9] to derive an upper bound on the sparsity for each column as

$$\mathbf{Pr}[\sum_{i=1}^m \mathbf{s}_{ij} \geq (1 + \epsilon)\gamma m] \leq \exp(-\frac{\epsilon^2\gamma m}{2(1+\epsilon/3)}).$$

for any $\epsilon > 0$. Union bounding over $L = \min(J, K)$ independent columns of $\mathbf{S}^{*(t)}$.

$$\mathbf{Pr}[\ \bigcup_{j=1}^L (\sum_{i=1}^m \mathbf{s}_{ij} \leq (1 + \epsilon)\gamma m)] \geq 1 - L\exp(-\frac{\epsilon^2\gamma m}{2(1+\epsilon/3)}).$$

Therefore, we conclude that if $m = \Omega(\log(L)/\gamma)$ then with probability $(1 - \delta_s)$ the expected number of non-zeros in a column of $\mathbf{S}^{*(t)}$ are $\mathcal{O}(\gamma m)$, where $\delta_s = L\exp(-\epsilon^2\gamma m/2(1 + \epsilon/3))$. ∎

**Lemma 7** *For any $\epsilon > 0$ suppose we have*

$$L \geq \frac{2}{(1-(1-\gamma)^m)\epsilon^2}\log(\frac{1}{\delta_p^{(t)}}),$$

*for $L = \min(J, K)$ and $\gamma = \alpha\beta$, then with probability at least $(1 - \delta_p)$,*

$$p = L(1 - (1 - \gamma)^m),$$

*where $\delta_p^{(t)} = \exp(-\frac{\epsilon^2}{2}L(1 - (1 - \gamma)^m))$.*

**Proof of Lemma 7** We begin by evaluating the probability that a column of $\mathbf{S}^{*(t)}$ has a non-zero element. Let $\mathbf{s}_{ij}$ be the indicator for the $(i, j)$ element of $\mathbf{S}^{*(t)}$ being non-zero, defined as

$$\mathbf{s}_{ij} = \mathbb{1}_{|\mathbf{S}_{ij}^*|>0}.$$

Further, let $w_j$ denote the number of non-zeros in the $j$-th column of $\mathbf{S}^{*(t)}$, defined as

$$w_j = \sum_{i=1}^m \mathbf{s}_{ij}.$$

Since each element of a column of $\mathbf{S}^{*(t)}$ is non-zero with probability $\gamma$, the probability that the $j$-th column of $\mathbf{S}^{*(t)}$ is an all zero vector is,

$$\mathbf{Pr}[w_j = 0] = (1 - \gamma)^m.$$

Therefore, the probability that the $j$-th column of $\mathbf{S}^{*(t)}$ has at least one non-zero element is given by

$$\mathbf{Pr}[w_j > 0] = 1 - (1-\gamma)^m. \tag{2}$$

Now, we are interested in the number of columns with at least one non-zero element among the $L = \min(J, K)$ independent columns of $\mathbf{S}^{*(t)}$, which we denote by $p$. Specifically, we analyze the following sum

$$p = \sum_{j=1}^{L} \mathbb{1}_{w_j > 0}.$$

Next, using (2) $\mathbf{E}[p] = L(1 - (1-\gamma)^m)$. Applying the result stated Lemma 13 (b),

$$\mathbf{Pr}[\sum_{j=1}^{L} \mathbb{1}_{w_j} \leq (1-\epsilon)\mathbf{E}[p]] \leq \exp(-\frac{\epsilon^2 E[p]}{2}) := \delta_p^{(t)}.$$

Therefore, if for any $\epsilon > 0$ we have

$$L \geq \frac{2}{(1-(1-\gamma)^m)\epsilon^2} \log(\frac{1}{\delta_p^{(t)}})$$

then with probability at least $(1 - \delta_p)$, $p = L(1 - (1-\gamma)^m)$, where $\delta_p^{(t)} = \exp(-\frac{\epsilon^2}{2} L(1 - (1-\gamma)^m))$. ∎

**Lemma 6** Suppose the input $\widehat{\mathbf{S}}^{(t)}$ to Alg. 2 is entry-wise $\zeta$ close to $\mathbf{S}^{*(t)}$, i.e., $|\widehat{\mathbf{S}}_{ij}^{(t)} - \mathbf{S}_{ij}^{*(t)}| \leq \zeta$ and has the correct signed-support as $\mathbf{S}^{*(t)}$. Then with probability at least $(1 - \delta_{\text{IHT}}^{(t)} - \delta_{\mathbf{B}_i}^{(t)})$, both $\widehat{\mathbf{B}}_i^{(t)}$ and $\widehat{\mathbf{C}}_i^{(t)}$ have the correct support, and $\left\| \frac{\mathbf{B}_i^{*(t)}}{\|\mathbf{B}_i^{*(t)}\|} - \pi_i \frac{\widehat{\mathbf{B}}_i^{(t)}}{\|\widehat{\mathbf{B}}_i^{(t)}\|} \right\| = \mathcal{O}(\zeta^2)$ and $\left\| \frac{\mathbf{C}_i^{*(t)}}{\|\mathbf{C}_i^{*(t)}\|} - \pi_i \frac{\widehat{\mathbf{C}}_i^{(t)}}{\|\widehat{\mathbf{C}}_i^{(t)}\|} \right\| = \mathcal{O}(\zeta^2)$, where $\delta_{\text{IHT}}^{(t)} = 2m \exp(-\frac{C^2}{\mathcal{O}^*(\epsilon_t^2)}) + 2s \exp(-\frac{1}{\mathcal{O}(\epsilon_t)})$ for $\|\mathbf{A}_i^{(t)} - \mathbf{A}_i^*\| \leq \epsilon_t$, and $\delta_{\mathbf{B}_i}^{(t)} = \exp(-\frac{\epsilon^2 J\alpha}{2(1+\epsilon/3)})$ for any $\epsilon > 0$.

**Proof of Lemma 6** The Iterative Hard Thresholding (IHT) results in an estimate of $\mathbf{X}^{*(t)}$ which has the correct signed support [11]. As a result, putting back the columns of $\widehat{\mathbf{X}}^{(t)}$ at the respective non-zero column locations of $\mathbf{Z}_1^{(t)\top}$, we arrive at the estimate $\widehat{\mathbf{S}}^{(t)}$ of $\mathbf{S}^{*(t)}$, which has the correct signed-support, we denote this estimate by $\widehat{\mathbf{S}}^{(t)}$. To recover the estimates $\widehat{\mathbf{B}}^{(t)}$ and $\widehat{\mathbf{C}}^{(t)}$, we use a SVD-based procedure. Specifically, we note that,

$$\mathbf{S}_{i,:}^{*(t)\top} = \mathbf{C}_i^{*(t)} \otimes \mathbf{B}_i^{*(t)} = vec(\mathbf{B}_i^{*(t)} \mathbf{C}_i^{*(t)\top})$$

As a result, the left and right singular vectors of the rank-1 matrix $\mathbf{B}_i^{*(t)} \mathbf{C}_i^{*(t)\top}$ are the columns $\mathbf{B}_i^{*(t)}$ and $\mathbf{C}_i^{*(t)}$, respectively (up to scaling).

Let $\mathbf{M}^{(i)}$ denote the $J \times K$ matrix formed by reshaping the vector $\widehat{\mathbf{S}}_{i,:}^{(t)\top}$. We choose the appropriately scaled left and right singular vectors corresponding to the largest singular value of $\mathbf{M}^{(i)}$ as our estimates $\widehat{\mathbf{B}}_i^{(t)}$ and $\widehat{\mathbf{C}}_i^{(t)}$, respectively.

First, notice that since $\widehat{\mathbf{S}}_{i,:}^{(t)\top}$ has the correct sign and support (due to Lemma 3), the support of matrix $\mathbf{M}^{(i)}$ is the same as $\mathbf{B}_i^{*(t)} \mathbf{C}_i^{*(t)\top}$. As a result, the estimates $\widehat{\mathbf{B}}_i^{(t)}$ and $\widehat{\mathbf{C}}_i^{(t)}$ have the correct support, and the error is only due to the scaling ambiguity on the support. This is due to the fact that the principal singular vectors ($\mathbf{u}$ and $\mathbf{v}$) align with the sparsity structure of $\mathbf{M}^{(i)}$ as they solve the following maximization problem also known as variational characterization of svd,

$$\sigma_1^2 = \max_{\|\mathbf{u}\|=1} \mathbf{u}^\top \mathbf{M}^{(i)} {\mathbf{M}^{(i)}}^\top \mathbf{u} = \max_{\|\mathbf{v}\|=1} \mathbf{v}^\top {\mathbf{M}^{(i)}}^\top \mathbf{M}^{(i)} \mathbf{v},$$

where $\sigma_1$ denotes the principal singular value. Therefore, since $\mathbf{M}^{(i)}$ has the correct sparsity structure as $\mathbf{B}_i^{*(t)} \mathbf{C}_i^{*(t)\top}$ the resulting $\mathbf{u}$ and $\mathbf{v}$ have the correct supports as well. Here,

$\mathbf{u}$ and $\mathbf{v}$ can be viewed as the normalized versions of $\widehat{\mathbf{B}}_i^{(t)}$ and $\widehat{\mathbf{C}}_i^{(t)}$, respectively, i.e., $\mathbf{u} = \widehat{\mathbf{B}}_i^{(t)}/\|\widehat{\mathbf{B}}_i^{(t)}\|$ and $\mathbf{v} = \widehat{\mathbf{C}}_i^{(t)}/\|\widehat{\mathbf{C}}_i^{(t)}\|$.

Let $\mathbf{E} = \mathbf{M}^{(i)} - \mathbf{B}_i^{*(t)}\mathbf{C}_i^{*(t)\top}$, now since $|\widehat{\mathbf{S}}_{ij}^{(t)} - \mathbf{S}_{ij}^{*(t)}| \leq \zeta$ and, from Lemma 3) $\widehat{\mathbf{S}}_{i,:}^{(t)}$ has the correct signed-support with probability $(1 - \delta_{\mathrm{IHT}}^{(t)})$, where $\delta_{\mathrm{IHT}}^{(t)} = 2m\exp(-\frac{C^2}{\mathcal{O}^*(\epsilon_t^2)}) + 2s\exp(-\frac{1}{\mathcal{O}(\epsilon_t)})$, and further using Claim 1, we have that the expected number of non-zeros in $\widehat{\mathbf{S}}_{i,:}^{(t)}$ are $JK\alpha\beta$, with probability at least $(1 - \delta_{\mathbf{B}_i}^{(t)})$, where $\delta_{\mathbf{B}_i}^{(t)} = \exp(-\frac{\epsilon^2 J\alpha}{2(1+\epsilon/3)})$ for some $\epsilon > 0$, we have

$$\|\mathbf{E}\| \leq \|\mathbf{E}\|_{\mathrm{F}} \leq \sqrt{JK\alpha\beta}\zeta,$$

Then, using the result in [14], and noting that $\sigma_1(\mathbf{B}_i^{(t)}\mathbf{C}_i^{(t)\top}) = \|\mathbf{B}_i^{(t)}\|\|\mathbf{C}_i^{(t)}\|$ and letting $\pi_i \in \{-1, 1\}$ (to resolve the sign ambiguity), we have that

$$\left\|\frac{\mathbf{B}_i^{*(t)}}{\|\mathbf{B}_i^{*(t)}\|} - \pi_i\mathbf{u}\right\| = \left\|\frac{\mathbf{B}_i^{*(t)}}{\|\mathbf{B}_i^{*(t)}\|} - \pi_i\frac{\widehat{\mathbf{B}}_i^{(t)}}{\|\widehat{\mathbf{B}}_i^{(t)}\|}\right\| \leq \frac{2^{3/2}(2\|\mathbf{B}_i^{(t)}\|\|\mathbf{C}_i^{(t)}\| + \sqrt{JK\alpha\beta}\zeta)\sqrt{JK\alpha\beta}\zeta}{\|\mathbf{B}_i^{(t)}\|^2\|\mathbf{C}_i^{(t)}\|^2}.$$

Next, since $\mathbf{E}[(\mathbf{B}_{ij}^{(t)})^2|(i,j) \in \mathrm{supp}(\mathbf{B}^{(t)})] = 1$ as per our distributional assumptions **Def.3**, we have

$\mathbf{E}[\|\mathbf{B}_{ji}^{*(t)}\|^2]$

$\quad = \mathbf{E}[(\mathbf{B}_{ji}^{*(t)})^2|(j,i) \in \mathrm{supp}(\mathbf{B}^{*(t)})]\mathbf{Pr}[(j,i) \in \mathrm{supp}(\mathbf{B}^{*(t)})] + 0.\mathbf{Pr}[(j,i) \notin \mathrm{supp}(\mathbf{B}^{*(t)})] = \alpha$

Similarly, $\mathbf{E}[\|\mathbf{C}_{ji}^{*(t)}\|^2] = \beta$. Substituting,

$$\left\|\frac{\mathbf{B}_i^{*(t)}}{\|\mathbf{B}_i^{*(t)}\|} - \pi_i\frac{\widehat{\mathbf{B}}_i^{(t)}}{\|\widehat{\mathbf{B}}_i^{(t)}\|}\right\| \leq \frac{2^{3/2}(2\sqrt{JK\alpha\beta} + \sqrt{JK\alpha\beta}\zeta)\sqrt{JK\alpha\beta}\zeta}{JK\alpha\beta} = \mathcal{O}(\zeta^2).$$

∎

**Claim 1** *Suppose $J = \Omega(\frac{1}{\alpha})$), then with probability at least $(1 - \delta_{\mathbf{B}_i}^{(t)})$,*

$$\sum_{j=1}^{JK}\mathrm{supp}(\mathbf{S}^*(i,j)) = JK\alpha\beta,$$

*where $\delta_{\mathbf{B}_i}^{(t)} = \exp(-\frac{\epsilon^2 J\alpha}{2(1+\epsilon/3)})$ for any $\epsilon > 0$.*

**Proof of Claim 1** In this lemma we establish an upper-bound on the number of non-zeros in a row of $\mathbf{S}^{*(t)}$. The $i$-th row of $\mathbf{S}^{*(t)}$ can be written as $\mathrm{vec}(\mathbf{B}_i^{*(t)}\mathbf{C}_i^{*(t)\top})$.

Since each element of matrix $\mathbf{B}^{*(t)}$ and $\mathbf{C}^{*(t)}$ are independently non-zero with probabilities $\alpha$ and $\beta$, the number of non-zeros in a column $\mathbf{B}_i^{*(t)}$ of $\mathbf{B}^{*(t)}$ are binomially distributed. Let $\mathbf{s}_j$ be the indicator for the $j$-th element of $\mathbf{B}_i^{*(t)}$ being non-zero, defined as

$$\mathbf{s}_i = \mathbb{1}_{|\mathbf{B}^{*(t)}(j,i)|>0}.$$

Then, the expected number of non-zeros (sparsity) in the $i$-th column of $\mathbf{B}^{*(t)}$ are given by

$$\mathbf{E}[\sum \mathrm{supp}(\mathbf{B}_i^{*(t)})] = \mathbf{E}[\sum_{j=1}^J \mathbf{s}_j] = J\alpha.$$

Since, $\alpha$ can be small, we use Lemma 13(a) [9] to derive an upper bound on the sparsity for each column as

$$\mathbf{Pr}[\sum_{j=1}^J \mathbf{s}_j \geq (1 + \epsilon)J\alpha] \leq \exp(-\frac{\epsilon^2 J\alpha}{2(1+\epsilon/3)}) := \delta_{\mathbf{B}_i}^{(t)}. \qquad (3)$$

for any $\epsilon > 0$.

Now we turn to the number of non-zeros in $\mathbf{S}_i^{*(t)} = \mathrm{vec}(\mathbf{B}_i^{*(t)}\mathbf{C}_i^{*(t)\top})$. We first note that the $j$-th column of $\mathbf{B}_i^{*(t)}\mathbf{C}_i^{*(t)\top}$ is given by $\mathbf{C}(j,i)^{*(t)}\mathbf{B}_i^{*(t)}$. This implies that the $j$-th column

can be all-zeros if $\mathbf{C}(j,i)^{*(t)} = 0$. As a result, the expected number of non-zeros in the $j$-th column of $\mathbf{B}_i^{*(t)}\mathbf{C}_i^{*(t)\top}$ can be written as,

$$\mathbf{E}[\sum \operatorname{supp}(\mathbf{C}_{ji}^{*(t)}\mathbf{B}_i^{*(t)})]$$
$$= \mathbf{E}[\sum \operatorname{supp}(\mathbf{C}_{ji}^{*(t)}\mathbf{B}_i^{*(t)})|\mathbf{C}_{ji}^{*(t)} \neq 0]\mathbf{Pr}[\mathbf{C}_{ji}^{*(t)} \neq 0]$$
$$+ \mathbf{E}[\sum \operatorname{supp}(\mathbf{C}_{ji}^{*(t)}\mathbf{B}_i^{*(t)})|\mathbf{C}_{ji}^{*(t)} = 0]\mathbf{Pr}[\mathbf{C}_{ji}^{*(t)} = 0]$$
$$= \mathbf{E}[\sum \operatorname{supp}(\mathbf{C}_{ji}^{*(t)}\mathbf{B}_i^{*(t)})|\mathbf{C}_{ji}^{*(t)} \neq 0]\mathbf{Pr}[\mathbf{C}_{ji}^{*(t)} \neq 0] = \mathbf{E}[\sum \operatorname{supp}(\mathbf{B}_i^{*(t)})]\mathbf{Pr}[\mathbf{C}_{ji}^{*(t)} \neq 0].$$

Now, from (3), we have that if we choose $J = \Omega(\frac{1}{\alpha})$) with probability at least $(1 - \delta_{\mathbf{B}_i}^{(t)})$, there are $J\alpha$ non-zeros in a column of $\mathbf{B}^{*(t)}$. Further since, $\mathbf{Pr}[\mathbf{C}_{ji}^{*(t)} \neq 0] = \beta$, we have that with probability at least $(1 - \delta_{\mathbf{B}_i}^{(t)})$,

$$\mathbf{E}[\sum \operatorname{supp}(\mathbf{C}_{ji}^{*(t)}\mathbf{B}_i^{*(t)})] = J\alpha\beta.$$

Furthermore, since there are $K$ columns in $\mathbf{B}_i^{*(t)}\mathbf{C}_i^{*(t)\top}$, with probability at least $(1 - \delta_{\mathbf{B}_i}^{(t)})$,

$$\mathbf{E}[\sum \operatorname{supp}(\operatorname{vec}(\mathbf{B}_i^{*(t)}\mathbf{C}_i^{*(t)\top}))] = \mathbf{E}[\sum_{j=1}^{JK} \operatorname{supp}(\mathbf{S}^{*(t)}(i,j))] = JK\alpha\beta.$$

$\blacksquare$

## D  Additional Theoretical Results

**Lemma 13** ***Relative Chernoff*** *[9] Let random variables $w_1, \ldots, w_\ell$ be independent, with $0 \leq w_i \leq 1$ for each $i$. Let $S_w = \sum_{i=1}^{\ell} w_i$, let $\nu = \mathbf{E}(S_w)$ and let $p = \nu/\ell$, then for any $\epsilon > 0$,*

$$(a) \quad \mathbf{Pr}[S_w - \nu \geq \epsilon\nu] \leq \exp(-\epsilon^2\nu/2(1 + \varepsilon/3)),$$
$$(b) \quad \mathbf{Pr}[S_w - \nu \leq \epsilon\nu] \leq \exp(-\epsilon^2\nu/2).$$

**Lemma 14 (Specialized Theorem 4 in [14] for singular vectors)** *Given $\mathbf{M}, \widetilde{\mathbf{M}} \in \mathbb{R}^{m \times n}$, where $\widetilde{\mathbf{M}} = \mathbf{M} + \mathbf{E}$ and the corresponding SVD of $\mathbf{M} = \mathbf{U}\boldsymbol{\Sigma}\mathbf{V}^\top$ and $\widetilde{\mathbf{M}} = \widehat{\mathbf{U}}\widetilde{\boldsymbol{\Sigma}}\widehat{\mathbf{V}}^\top$, the sine of angle between the principal left (and right) singular vectors of matrices $\mathbf{M}$ and $\widetilde{\mathbf{M}}$ is given by*

$$\sin \Theta(\mathbf{U}_1, \widetilde{\mathbf{U}}_1) \leq \frac{2(2\sigma_1 + \|\mathbf{E}\|_2)(\min(\|\mathbf{E}\|_2, \|\mathbf{E}\|_\mathrm{F}))}{\sigma_1^2},$$

*where $\sigma_1$ is the principal singular value corresponding to $\mathbf{U}_1$. Furthermore, there exists $\pi \in -1, 1$ such that*

$$\|\mathbf{U}_1 - \pi\widetilde{\mathbf{U}}_1\| \leq \frac{2^{3/2}(2\sigma_1 + \|\mathbf{E}\|_2)(\min(\|\mathbf{E}\|_2, \|\mathbf{E}\|_\mathrm{F}))}{\sigma_1^2}.$$

**Theorem 1 ([11])** *Suppose that assumptions A.1-A.6 hold, and Alg. 1 is provided with $p = \widetilde{\Omega}(mk^2)$ new samples generated according to model (1) at each iteration $t$. Then for some $0 < \omega < 1/2$, the estimate $\mathbf{A}^{(t)}$ at $(t)$-th iteration satisfies*

$$\|\mathbf{A}_i^{(t)} - \mathbf{A}_i^*\|^2 \leq (1 - \omega)^t\|\mathbf{A}_i^{(0)} - \mathbf{A}_i^*\|^2, \text{ for all } t = 1, 2, \ldots.$$

*Furthermore, given $R = \Omega(\log(n))$, with probability at least $(1 - \delta_{alg}^{(t)})$ for some small constant $\delta_{alg}^{(t)}$, the coefficient estimate $\widehat{\mathbf{x}}_i^{(t)}$ at $t$-th iteration has the correct signed-support and satisfies*

$$(\widehat{\mathbf{x}}_i^{(t)} - \mathbf{x}_i^*)^2 = \mathcal{O}(k(1 - \omega)^{t/2}\|\mathbf{A}_i^{(0)} - \mathbf{A}_i^*\|), \text{ for all } i \in \operatorname{supp}(\mathbf{x}^*).$$

**Table E.1:** Tensor factorization results $\alpha, \beta = 0.005$ averaged across 3 trials. Here, $T(\text{supp}(\widehat{\mathbf{X}}^{(T)})?)$ field shows the number of iterations $T$ to reach the target tolerance, while the categorical field, $\text{supp}(\widehat{\mathbf{X}}^{(T)})$ indicates if the support of the recovered $\widehat{\mathbf{X}}^{(T)}$ matches that of $\mathbf{X}^{*(T)}$ (Y) or not (N).

| $(J,K)$ | Method | $m=50$ $\frac{\|\mathbf{A}^*-\mathbf{A}^{(T)}\|_F}{\|\mathbf{A}^*\|_F}$ | $\frac{\|\mathbf{X}^{*(T)}-\mathbf{X}^{(T)}\|_F}{\|\mathbf{X}^{*(T)}\|_F}$ | $T(\text{supp}(\widehat{\mathbf{x}})?)$ | $m=150$ $\frac{\|\mathbf{A}^*-\mathbf{A}^{(T)}\|_F}{\|\mathbf{A}^*\|_F}$ | $\frac{\|\mathbf{X}^{*(T)}-\mathbf{X}^{(T)}\|_F}{\|\mathbf{X}^{*(T)}\|_F}$ | $T(\text{supp}(\widehat{\mathbf{x}})?)$ | $m=300$ $\frac{\|\mathbf{A}^*-\mathbf{A}^{(T)}\|_F}{\|\mathbf{A}^*\|_F}$ | $\frac{\|\mathbf{X}^{*(T)}-\mathbf{X}^{(T)}\|_F}{\|\mathbf{X}^{*(T)}\|_F}$ | $T(\text{supp}(\widehat{\mathbf{x}})?)$ |
|---|---|---|---|---|---|---|---|---|---|---|
| **100** | NOODL | 5.38e-11 | 2.38e-16 | 245 (Y) | 7.04e-11 | 2.24e-16 | 257 (Y) | 5.48e-11 | 5.14e-13 | 240 (Y) |
|  | Arora(b) | 1.87e-06 | 1.14e-05 | 245 (N) | 2.09e-03 | 1.41e-03 | 257 (N) | 2.70e-03 | 2.41e-03 | 240 (N) |
|  | Arora(u) | 6.78e-08 | 1.14e-05 | 245 (N) | 8.94e-05 | 7.38e-05 | 257 (N) | 1.72e-04 | 8.76e-05 | 240 (N)) |
|  | Mairal | 4.40e-03 | 2.00e-03 | 245 (N) | 4.90e-03 | 6.87e-03 | 257 (N) | 6.00e-03 | 5.10e-03 | 240 (N) |
| **300** | NOODL | 5.72e-11 | 1.13e-12 | 61 (Y) | 6.74e-11 | 5.44e-13 | 89 (Y) | 9.10e-11 | 1.27e-12 | 168 (Y) |
|  | Arora(b) | 2.13e-03 | 2.86e-03 | 61 (N) | 5.90e-04 | 4.50e-04 | 89 (N) | 1.00e-03 | 1.10e-03 | 168 (N) |
|  | Arora(u) | 2.04e-04 | 2.70e-04 | 61 (N) | 3.82e-05 | 4.26e-05 | 89 (N) | 1.04e-04 | 1.09e-04 | 168 (N) |
|  | Mairal | 2.05e-01 | 2.28e-01 | 61 (N) | 1.19e-02 | 1.09e-02 | 89 (N) | 1.07e-02 | 8.40e-03 | 168 (N) |
| **500** | NOODL | 5.49e-11 | 2.34e-16 | 50 (Y) | 8.15e-11 | 1.25e-12 | 76 (Y) | 9.27e-11 | 1.41e-12 | 160 (Y) |
|  | Arora(b) | 1.11e-04 | 1.34e-04 | 50 (N) | 5.75e-04 | 5.60e-04 | 76 (N) | 6.32e-04 | 2.71e-03 | 160 (N) |
|  | Arora(u) | 9.75e-06 | 1.50e-05 | 50 (N) | 4.30e-05 | 4.73e-05 | 76 (N) | 5.55e-05 | 2.28e-03 | 160 (N) |
|  | Mairal | 1.23e-01 | 1.10e-01 | 50 (N) | 1.73e-02 | 1.20e-02 | 76 (N) | 1.44e-02 | 5.99e-02 | 160 (N) |

| $(J,K)$ | Method | $m=450$ $\frac{\|\mathbf{A}^*-\mathbf{A}^{(T)}\|_F}{\|\mathbf{A}^*\|_F}$ | $\frac{\|\mathbf{X}^{*(T)}-\mathbf{X}^{(T)}\|_F}{\|\mathbf{X}^{*(T)}\|_F}$ | $T(\text{supp}(\widehat{\mathbf{x}})?)$ | $m=500$ $\frac{\|\mathbf{A}^*-\mathbf{A}^{(T)}\|_F}{\|\mathbf{A}^*\|_F}$ | $\frac{\|\mathbf{X}^{*(T)}-\mathbf{X}^{(T)}\|_F}{\|\mathbf{X}^{*(T)}\|_F}$ | $T(\text{supp}(\widehat{\mathbf{x}}))$ |
|---|---|---|---|---|---|---|---|
| **100** | NOODL | 7.82e-11 | 1.79e-12 | 257 (Y) | 8.30e-11 | 6.39e-13 | 300 (Y) |
|  | Arora(b) | 3.80e-03 | 3.20e-03 | 257 (N) | 2.80e-03 | 3.06e-03 | 300 (N) |
|  | Arora(u) | 3.06e-04 | 1.82e-04 | 257 (N) | 2.52e-04 | 2.76e-04 | 300 (N) |
|  | Mairal | 7.20e-03 | 6.90e-03 | 257 (N) | 8.27e-03 | 8.07e-03 | 300 (N) |
| **300** | NOODL | 9.43e-11 | 1.56e-12 | 201 (Y) | 9.50e-11 | 1.63e-12 | 265 (Y) |
|  | Arora(b) | 9.77e-04 | 1.04e-03 | 201 (N) | 1.03e-03 | 9.36e-04 | 265 (N) |
|  | Arora(u) | 1.42e-04 | 1.68e-04 | 201 (N) | 1.27e-04 | 1.23e-04 | 265 (N) |
|  | Mairal | 1.47e-02 | 1.39e-02 | 201 (N) | 9.40e-03 | 1.05e-02 | 265 (N) |
| **500** | NOODL | 9.77e-11 | 1.60e-12 | 196 (Y) | 9.72e-11 | 1.84e-12 | 264 (Y) |
|  | Arora(b) | 5.99e-04 | 5.30e-03 | 196 (N) | 6.04e-04 | 6.37e-03 | 264 (N) |
|  | Arora(u) | 5.91e-05 | 5.30e-03 | 196 (N | 8.08e-05 | 6.37e-03 | 264 (N) |
|  | Mairal | 3.22e-01 | 2.87e-01 | 196 (N) | 2.46e-02 | 1.70e-01 | 264 (N) |

# E  Experimental Evaluation

We now detail the specifics of the experiments and present additional results corresponding to section 5 for synthetic data experiments and real-world data experiments, respectively.

**Distributed Implementations**: Since the updates of $\mathbf{X}^{(r)(t)}$ columns are independent of each other, `TensorNOODL` is amenable for large-scale implementation in highly distributed settings. As a result, it is especially suitable for handling the tensor decomposition applications. Furthermore, the online nature of `TensorNOODL` allows for life-long learning.

**Note on Initialization**: For synthetic data simulations, since the ground-truth factors are known, we can initialize the dictionary factor such that the requirements of Def. 1 are met. In real-world data setting, the ground-truth is unknown and our initialization requirement can be met by existing algorithms, such as [1]. Consequently, in real-world experiments we use [1] to initialize the dictionary factor $\mathbf{A}^{(0)}$. Here, we run the initialization algorithm and communicate the estimate $\mathbf{A}^{(0)}$ to each worker at the beginning of the distributed operation.

## E.1  Synthetic Data Simulations

### E.1.1  Experimental Set-up

**Overview of Experiments** – As discussed in section 5, we analyze the performance of the algorithm across different choices of tensor dimensions $(J, K)$ for a fixed $n = 300$, its rank$(m)$ and the sparsity of factors $\mathbf{B}^{*(t)}$ and $\mathbf{C}^{*(t)}$ controlled by parameters $(\alpha, \beta)$, for recovery of the constituent factors using three Monte-Carlo runs. For each of these runs, we analyze the recovery performance across three choices of dimensions $J = K = \{100, 300, 500\}$, five choices of rank $m = \{50, 150, 300, 450, 600\}$, and three choices of the sparsity parameters $\alpha = \beta = \{0.005, 0.01, 0.05\}$. The simulation results corresponding to $\alpha = \beta = \{0.005, 0.01, 0.05\}$ are shown in Table E.1, E.2, and E.4, respectively.

We compare `TensorNOODL` with related techniques which are also agnostic to the tensor structure for fairness. Specifically, we compare `TensorNOODL` with online dictionary learning algorithms presented in [1] (`Arora(b)` (incurs bias) and `Arora(u)` (claim no bias))[1], and [8],

**Table E.2:** Tensor factorization results $\alpha, \beta = 0.01$ averaged across 3 trials. Here, $T(\text{supp}(\widehat{\mathbf{X}}^{(T)})?)$ field shows the number of iterations $T$ to reach the target tolerance, while the categorical field, $\text{supp}(\widehat{\mathbf{X}}^{(T)})$ indicates if the support of the recovered $\widehat{\mathbf{X}}^{(T)}$ matches that of $\mathbf{X}^{*(T)}$ (Y) or not (N).

| $(J,K)$ | Method | $m=50$ | | | $m=150$ | | | $m=300$ | | |
|---|---|---|---|---|---|---|---|---|---|---|
| | | $\frac{\|\mathbf{A}^*-\mathbf{A}^{(T)}\|_F}{\|\mathbf{A}^*\|_F}$ | $\frac{\|\mathbf{X}^{*(T)}-\mathbf{X}^{(T)}\|_F}{\|\mathbf{X}^{*(T)}\|_F}$ | $T(\text{supp}(\widehat{\mathbf{X}})?)$ | $\frac{\|\mathbf{A}^*-\mathbf{A}^{(T)}\|_F}{\|\mathbf{A}^*\|_F}$ | $\frac{\|\mathbf{X}^{*(T)}-\mathbf{X}^{(T)}\|_F}{\|\mathbf{X}^{*(T)}\|_F}$ | $T(\text{supp}(\widehat{\mathbf{X}})?)$ | $\frac{\|\mathbf{A}^*-\mathbf{A}^{(T)}\|_F}{\|\mathbf{A}^*\|_F}$ | $\frac{\|\mathbf{X}^{*(T)}-\mathbf{X}^{(T)}\|_F}{\|\mathbf{X}^{*(T)}\|_F}$ | $T(\text{supp}(\widehat{\mathbf{X}})?)$ |
| 100 | NOODL | 5.50e-11 | 5.66e-13 | 91 (Y) | 7.59e-11 | 5.28e-13 | 112 (Y) | 4.34e-11 | 1.62e-12 | 190 (Y) |
| | Arora(b) | 3.93e-03 | 5.80e-03 | 91 (N) | 2.61e-03 | 1.58e-03 | 112 (N) | 2.70e-03 | 3.00e-03 | 190 (N) |
| | Arora(u) | 4.35e-04 | 6.77e-04 | 91 (N) | 6.87e-04 | 1.05e-04 | 112 (N) | 2.98e-04 | 3.04e-04 | 190 (N) |
| | Mairal | 4.03e-02 | 1.26e-02 | 91 (N) | 1.34e-02 | 1.25e-02 | 112 (N) | 1.18e-02 | 1.25e-02 | 190 (N) |
| 300 | NOODL | 6.78e-11 | 5.75e-13 | 51 (Y) | 6.35e-11 | 1.54e-12 | 76 (Y) | 8.64e-11 | 2.06e-12 | 158 (Y) |
| | Arora(b) | 4.08e-04 | 4.76e-04 | 51 (N) | 1.03e-03 | 1.08e-03 | 76 (N) | 1.04e-03 | 1.17e-02 | 158 (N) |
| | Arora(u) | 1.99e-05 | 1.46e-05 | 51 (N) | 1.03e-04 | 9.59e-05 | 76 (N) | 2.17e-04 | 1.17e-02 | 158 (N) |
| | Mairal | 1.64e-01 | 1.63e-01 | 51 (N) | 2.61e-02 | 2.64e-02 | 76 (N) | 2.81e-02 | 1.58e-01 | 158 (N) |
| 500 | NOODL | 6.92e-11 | 8.78e-13 | 46 (Y) | 8.77e-11 | 1.77e-12 | 77 (Y) | 9.35e-11 | 2.12e-12 | 156 (Y) |
| | Arora(b) | 3.48e-04 | 3.28e-04 | 46 (N) | 5.42e-04 | 6.40e-03 | 77 (N) | 5.69e-04 | 2.41e-03 | 156 (N) |
| | Arora(u) | 2.56e-05 | 3.70e-05 | 46 (N) | 4.81e-05 | 6.40e-03 | 77 (N) | 1.08e-04 | 9.30e-03 | 156 ((N) |
| | Mairal | 1.56e-01 | 1.53e-01 | 46 (N) | 5.28e-02 | 1.30e-01 | 77 (N) | 2.53e-02 | 1.57e-01 | 156 (N) |

| $(J,K)$ | Method | $m=450$ | | | $m=500$ | | |
|---|---|---|---|---|---|---|---|
| | | $\frac{\|\mathbf{A}^*-\mathbf{A}^{(T)}\|_F}{\|\mathbf{A}^*\|_F}$ | $\frac{\|\mathbf{X}^{*(T)}-\mathbf{X}^{(T)}\|_F}{\|\mathbf{X}^{*(T)}\|_F}$ | $T(\text{supp}(\widehat{\mathbf{X}})?)$ | $\frac{\|\mathbf{A}^*-\mathbf{A}^{(T)}\|_F}{\|\mathbf{A}^*\|_F}$ | $\frac{\|\mathbf{X}^{*(T)}-\mathbf{X}^{(T)}\|_F}{\|\mathbf{X}^{*(T)}\|_F}$ | $T(\text{supp}(\widehat{\mathbf{X}})?)$ |
| 100 | NOODL | 9.48e-11 | 1.78e-12 | 211 (Y) | 7.27e-11 | 1.94e-12 | 279 (Y) |
| | Arora(b) | 3.30e-03 | 4.00e-03 | 211 (N) | 3.40e-03 | 3.37e-03 | 279 (N) |
| | Arora(u) | 8.55e-04 | 1.27e-03 | 211 (N) | 6.83e-04 | 6.49e-04 | 279 (N) |
| | Mairal | 8.00e-03 | 6.60e-03 | 211 (N) | 8.77e-03 | 9.93e-03 | 279 (N) |
| 300 | NOODL | 9.43e-11 | 2.92e-12 | 192 (Y) | 9.33e-11 | 2.54e-12 | 252 (Y) |
| | Arora(b) | 1.00e-03 | 1.25e-02 | 192 (N) | 1.13e-03 | 1.54e-02 | 252 (N) |
| | Arora(u) | 2.22e-04 | 1.25e-02 | 192 (N) | 2.69e-04 | 1.54e-02 | 252 (N) |
| | Mairal | 1.39e-01 | 2.03e-01 | 192 (N) | 1.92e-02 | 1.83e-01 | 252 (N) |
| 500 | NOODL | 9.60e-11 | 2.41e-12 | 186 (Y) | 9.82e-11 | 2.66e-12 | 249 (Y) |
| | Arora(b) | 6.49e-04 | 1.20e-02 | 186 (N) | 6.55e-04 | 1.42e-02 | 249 (N) |
| | Arora(u) | 1.39e-04 | 1.20e-02 | 186 (N) | 1.55e-04 | 1.42e-02 | 249 (N) |
| | Mairal | 6.38e-02 | 1.54e-01 | 186 (N) | 1.74e-02 | 1.79e-01 | 249 (N) |

which can be viewed as a variant of $\ell_1$-regularized Alternating Least Squares (ALS), taking the matrix factorization view of the tensor decomposition task.

**Data Generation** – For synthetic data experiments, we circumvent using the computationally expensive initialization algorithm of [1] (running time of $\widetilde{\mathcal{O}}(m^2n^2s)$) by providing all algorithms with perturbed ground-truth dictionary (in accordance with **A.2**). Specifically, for each experiment we draw entries of the dictionary factor matrix $\mathbf{A}^* \in \mathbb{R}^{n \times m}$ from $\mathcal{N}(0,1)$, and normalize its columns to be unit-norm. To form $\mathbf{A}^{(0)}$ in accordance with **A.2**, we perturb $\mathbf{A}^*$ with random Gaussian noise and normalized its columns, such that it is column-wise $2/\log(n)$ away from $\mathbf{A}^*$ in $\ell_2$ norm sense. To form the sparse factors $\mathbf{B}^{*(t)}$ and $\mathbf{C}^{*(t)}$, we assign their entries to the support independently with probability $\alpha$ and $\beta$, respectively, and then draw the values on the support from the Rademacher distribution.

**Parameters Setting**: We set `TensorNOODL` specific IHT parameters $\eta_x = 0.2$ and $\tau = 0.1$ for all experiments. As recommended by our main result, the dictionary step-size parameter $\eta_A$ is set proportional to $m/k$. Since `TensorNOODL`, `Arora(b)`, and `Arora(u)` all rely on an approximate gradient descent strategy for dictionary update, we use the same step-size $\eta_A$ for a fair comparison depending upon the choice of rank $m$, and probabilities $(\alpha, \beta)$ as per **A.5**; Table **E.3** lists the step-size choices. Here, `Mairal` does not employ such a parameter.

**Evaluation Metrics**: We run all algorithms till one of them achieves target tolerance (error in the factor $\mathbf{A}$, $\epsilon_T$) of $10^{-10}$, and report the number of iterations $T$ for each experiment.

we find that given equal number of samples, the `Arora(u)` is only slightly better than `Arora(b)`; see Fig.5 for the convergence results.

**Table E.3:** Choosing the step-size $(\eta_A)$ for the dictionary update step. We use the same dictionary update step-size parameter $(\eta_A)$ for `TensorNOODL`, `Arora(b)`, and `Arora(u)` depending upon the choice of rank $m$, and probabilities $(\alpha, \beta)$, as per **A.5**.

| Rank $(m)$ | Step-size $(\eta_A)$ | Notes |
|---|---|---|
| 50 | 20 | For $(\alpha, \beta) = 0.005$, we use $\eta_A = 5$ |
| 150 | 40 | – |
| 300 | 40 | – |
| 450 | 50 | – |
| 600 | 50 | – |

**Table E.4:** Tensor factorization results $\alpha, \beta = 0.05$ averaged across 3 trials. Here, $T(\mathrm{supp}(\widehat{\mathbf{X}}^{(T)})?)$ field shows the number of iterations $T$ to reach the target tolerance, while the categorical field, $\mathrm{supp}(\widehat{\mathbf{X}}^{(T)})$ indicates if the support of the recovered $\widehat{\mathbf{X}}^{(T)}$ matches that of $\mathbf{X}^{*(T)}$ (Y) or not (N).

| $(J,K)$ | Method | $m=50$ | | | $m=150$ | | | $m=300$ | | |
|---|---|---|---|---|---|---|---|---|---|---|
| | | $\frac{\|\mathbf{A}^*-\mathbf{A}^{(T)}\|_F}{\|\mathbf{A}^*\|_F}$ | $\frac{\|\mathbf{X}^{*(T)}-\mathbf{X}^{(T)}\|_F}{\|\mathbf{X}^{*(T)}\|_F}$ | $T(\mathrm{supp}(\widehat{\mathbf{X}})?)$ | $\frac{\|\mathbf{A}^*-\mathbf{A}^{(T)}\|_F}{\|\mathbf{A}^*\|_F}$ | $\frac{\|\mathbf{X}^{*(T)}-\mathbf{X}^{(T)}\|_F}{\|\mathbf{X}^{*(T)}\|_F}$ | $T(\mathrm{supp}(\widehat{\mathbf{X}})?)$ | $\frac{\|\mathbf{A}^*-\mathbf{A}^{(T)}\|_F}{\|\mathbf{A}^*\|_F}$ | $\frac{\|\mathbf{X}^{*(T)}-\mathbf{X}^{(T)}\|_F}{\|\mathbf{X}^{*(T)}\|_F}$ | $T(\mathrm{supp}(\widehat{\mathbf{X}})?)$ |
| **100** | NOODL | 8.03e-11 | 3.17e-12 | 46 (Y) | 7.71e-11 | 4.92e-12 | 63 (Y) | 9.66e-11 | 6.01e-12 | 110 (Y) |
| | Arora(b) | 2.90e-03 | 3.00e-03 | 46 (N) | 4.60e-03 | 3.39e-02 | 63 (N) | 5.50e-03 | 4.89e-02 | 110 (N) |
| | Arora(u) | 8.97e-04 | 8.48e-04 | 46 (N) | 1.90e-03 | 3.40e-02 | 63 (N) | 2.80e-03 | 4.90e-02 | 110 (N) |
| | Mairal | 1.57e-01 | 1.67e-01 | 46 (N) | 3.63e-02 | 1.54e-01 | 63 (N) | 2.32e-02 | 1.99e-01 | 110 (N) |
| **300** | NOODL | 6.51e-11 | 3.27e-12 | 42 (Y) | 9.05e-11 | 5.61e-12 | 60 (Y) | 9.10e-11 | 7.01e-12 | 107 (Y) |
| | Arora(b) | 1.40e-03 | 1.95e-02 | 42 (N) | 2.50e-03 | 3.55e-02 | 60 (N) | 3.20e-03 | 5.04e-02 | 107 (N) |
| | Arora(u) | 2.48e-04 | 1.95e-02 | 42 (N) | 6.35e-04 | 3.56e-02 | 60 (N) | 9.48e-04 | 5.05e-02 | 107 (N) |
| | Mairal | 6.24e-02 | 1.11e-01 | 42 (N) | 3.05e-02 | 1.59e-01 | 60(N) | 1.91e-02 | 2.09e-01 | 107 (N) |
| **500** | NOODL | 7.72e-11 | 3.86e-12 | 42 (Y) | 8.44e-11 | 5.63e-12 | 59 (Y) | 9.64e-11 | 7.34e-12 | 106 (Y) |
| | Arora(b) | 1.30e-03 | 2.02e-02 | 42 (N) | 2.10e-03 | 3.55e-02 | 59 (N) | 2.80e-03 | 5.03e-02 | 106 (N) |
| | Arora(u) | 1.39e-04 | 2.02e-02 | 42 (N) | 3.82e-04 | 3.56e-02 | 59 (N) | 5.66e-04 | 5.05e-02 | 106 (N) |
| | Mairal | 6.12e-02 | 1.10e-01 | 42 (N) | 2.93e-02 | 1.58e-01 | 59 (N) | 1.80e-02 | 2.11e-01 | 106 (N) |

| $(J,K)$ | Method | $m=450$ | | | $m=500$ | | |
|---|---|---|---|---|---|---|---|
| | | $\frac{\|\mathbf{A}^*-\mathbf{A}^{(T)}\|_F}{\|\mathbf{A}^*\|_F}$ | $\frac{\|\mathbf{X}^{*(T)}-\mathbf{X}^{(T)}\|_F}{\|\mathbf{X}^{*(T)}\|_F}$ | $T(\mathrm{supp}(\widehat{\mathbf{X}})?)$ | $\frac{\|\mathbf{A}^*-\mathbf{A}^{(T)}\|_F}{\|\mathbf{A}^*\|_F}$ | $\frac{\|\mathbf{X}^{*(T)}-\mathbf{X}^{(T)}\|_F}{\|\mathbf{X}^{*(T)}\|_F}$ | $T(\mathrm{supp}(\widehat{\mathbf{X}})?)$ |
| **100** | NOODL | 8.92e-11 | 7.29e-12 | 115 (Y) | 8.71e-11 | 1.06e-11 | 131 (Y) |
| | Arora(b) | 7.50e-03 | 6.17e-02 | 115 (N) | 9.16e-03 | 7.36e-02 | 131 (N) |
| | Arora(u) | 4.40e-03 | 6.19e-02 | 115 (N) | 5.70e-03 | 7.40e-02 | 131 (N) |
| | Mairal | 8.79e-02 | 2.27e-01 | 115 (N) | 2.81e-02 | 2.56e-01 | 131 (N) |
| **300** | NOODL | 9.20e-11 | 8.41-12 | 110 (Y) | 8.49e-11 | 9.03e-12 | 128 (Y) |
| | Arora(b) | 4.00e-03 | 6.16e-02 | 110 (N) | 4.90e-03 | 7.39e-02 | 128 (N) |
| | Arora(u) | 1.40e-03 | 6.18e-02 | 110 (N) | 1.83e-03 | 7.42e-02 | 128 (N) |
| | Mairal | 4.85e-02 | 2.19e-01 | 110 (N) | 2.32e-02 | 2.63e-01 | 128 (N) |
| **500** | NOODL | 8.95e-11 | 8.21e-12 | 109 (Y) | 9.06e-11 | 9.29e-12 | 127 (Y) |
| | Arora(b) | 3.60e-03 | 6.21e-02 | 109 (N) | 4.40e-03 | 7.40e-02 | 127 (N) |
| | Arora(u) | 8.54e-04 | 6.23e-02 | 109 (N) | 1.10e-03 | 7.44e-02 | 127 (N) |
| | Mairal | 4.62e-02 | 2.20e-01 | 109 (N) | 4.05e-02 | 2.56e-01 | 127 (N) |

Note that, in all cases `TensorNOODL` achieves the tolerance first, and in some cases with the algorithms considered in the analysis. Next, since recovery of $\mathbf{A}^*$ and $\mathbf{X}^{*(t)}$ is vital for the success of the tensor factorization task, we report the relative Frobenius error for each of these matrices, i.e., for a recovered matrix $\widehat{\mathbf{M}}$, we report $\|\widehat{\mathbf{M}} - \mathbf{M}^*\|_F/\|\mathbf{M}^*\|_F$. In addition, since the dictionary learning task focuses on recovering the sparse matrix $\mathbf{X}^{*(t)}$, it is agnostic to the transposed Khatri-Rao structure $\mathbf{S}^{*(t)}$. As a result, for recovering the sparse factors $\mathbf{B}^{*(t)}$ and $\mathbf{C}^{*(t)}$ is crucial for exact support recovery of $\mathbf{X}^{*(t)}$. Therefore, we report if the support has been exactly recovered or not.

### E.1.2  Other Considerations

**Reproducible Results**: The code employed is made available as part of the supplementary material. We fix the random seeds (to $42, 26$, and $91$) for each Monte Carlo run to ensure reproducibility of the results shown in this work. The experiments were run on a HP Haswell Linux Cluster. The processing of data samples for the sparse coefficients ($\widehat{\mathbf{X}}^{*(t)}$) was split across 20 workers (cores), allocated a total of 200 GB RAM. For `Arora(b)`, `Arora(u)`, and `Mairal`, the coefficient recovery was switched between Fast Iterative Shrinkage-Thresholding Algorithm (FISTA) [3], an accelerated proximal gradient descent algorithm, or a stochastic-version of Iterative Shrinkage-Thresholding Algorithm (ISTA) [4, 5] depending upon the size of the data samples available for learning (see the discussion of the coefficient update step below); see also [3] for details.

**Sparse Factor Recovery Considerations**: In [1], the authors present two algorithms – a simple algorithm with a sample complexity of $\widetilde{\Omega}(ms)$ which incurs an estimation bias (`Arora(b)`), and a more involved variant for unbiased estimation of the dictionary whose sample complexity was not established `Arora(u)`. However, these algorithms do not provide guarantees on, or recover the sparse coefficients. As a result, we need to adopt an additional $\ell_1$ minimization based coefficient recovery step. Further, the algorithm proposed by [8] can be viewed as a variant of regularized alternating least squares algorithm which employs $\ell_1$ regularization for the recovery of the transposed Khatri-Rao structured matrix.

To form the coefficient estimates for `Arora(b)`, `Arora(u)`, and `Mairal` '09 we solve the Lasso [12] program using a stochastic-version of Iterative Shrinkage-Thresholding Algorithm (ISTA) [4, 5] (or Fast Iterative Shrinkage-Thresholding Algorithm (FISTA) [3] if $p$ is small) and report the best estimate (in terms of relative Frobenius error) across 10 values of

the regularization parameter. The stochastic projected gradient descent is necessary to make coefficient recovery tractable since size of $\mathbf{X}^{*(t)}$ grows quickly with $(\alpha, \beta)$. For these algorithms, coefficient estimation step the slowest step since it has to scan through different values of the regularization parameters to arrive at an estimate. In contrast, `TensorNOODL` does not require such an expensive tuning procedure, while providing recovery guarantees on the recovered coefficients.

Note that in practice ISTA and FISTA can be parallelized as well, but tuning of the regularization parameters still involves (an expensive) grid search. Arguably even if each step of these algorithms (ISTA and FISTA) take the same amount of time as that of `TensorNOODL`, the search over, say 10, values of the regularization parameters will still be take 10 times the time. As a result, `TensorNOODL` is an attractive choice as it does not involve an expensive tuning procedure.

**Discussion**: Table E.1, E.2, and E.4 show the results of the analysis averaged across the three Monte Carlo runs, for $\alpha = \beta = \{0.005, 0.01, 0.05\}$, respectively. We note that for every choice of $(J, K)$, $m$, and $(\alpha, \beta)$, `TensorNOODL` is orders of magnitude superior to related techniques. In addition, it also recovers the support correctly in all of the cases, ensuring that the sparse factors can be recovered correctly. Specifically, the sparse factors $\mathbf{B}^{*(t)}$ and $\mathbf{C}^{*(t)}$ can be recovered (up to permutation and scaling) via Alg. 2.

In addition, our result also shows that for the given task, where a number of mode-1 fibers are zero, processing only the non-zero fibers may lead to significant gains since there is no need to solve large sparse approximation sub-problems as is the case with ALS (which also requires additional tuning). Therefore, it seems that leveraging tensor structure for this our model may increase the computational complexity. Nevertheless, the tensor structure can potentially be useful in presence of noise, where this structure is not obvious.

## E.2   Real-world Data Simulations

### E.2.1   Analysis of the Enron Dataset

**Enron Email Dataset**: Sparsity-regularized ALS-based tensor factorization techniques, albeit possessing limited convergence guarantees, have been a popular choice to analyze the Enron Email Dataset ($184 \times 184 \times 44$) [7, 2]. We now use `TensorNOODL` to analyze the email activity of 184 Enron employees over 44 weeks (Nov. '98 –Jan. '02) during the period before and after the financial irregularities were uncovered.

The Enron Email Dataset ($184 \times 184 \times 44$) consists of email exchanges between 184 employees over 44 weeks (Nov. '98 –Jan. '02) which includes the period before and after the financial irregularities were uncovered. In general, every person in an organization (like Enron) communicates with only a subset of employees, as a result the tensor of email activity (Employees vs. Employees vs. Time) naturally has the model analyzed in this work. Moreover, as pointed out by [6] "...*in* 2000 *Enron had a segmented culture with directives being sent from on-high and sporadic feedback*". Meaning that different units within the organization exhibited clustered communication structure. This motivates us to analyze the dataset for the presence characteristic ways of communications between different business units.

We run `TensorNOODL` in batch setting here, this is to showcase that in practice `TensorNOODL` also works in batch settings, and also to overcome the limited size of the Enron Dataset.

**Data Preparation and Parameters**: For `TensorNOODL` and `Mairal '09`, we use the initialization algorithm of [1], which yielded 4 dictionary elements. Following this, we use these techniques in batch setting to simultaneously identify email activity patterns and cluster employees. We also compare our results to [7], which just aims to cluster the employees by imposing sparsity constraint on one of the factors, and does not learn the patterns. As opposed to [7], `TensorNOODL` did not require us to guess the number of dictionary elements to be used. We use Alg. 2 to identify the employees corresponding to email activity patterns from the recovered sparse factors.

As in case of [7], we transform each non-zero element $\underline{\mathbf{Z}}(i, j, k)^{(t)}$ of the dataset as

$$\underline{\mathbf{Z}}(i, j, k)^{(t)} = \log_2(\underline{\mathbf{Z}}(i, j, k)) + 1,$$

(a)

(b)

(c) Cluster Quality: False Positives/ Cluster Size

| Method | Legal | Pipeline | Executive | Trading |
|--------|-------|----------|-----------|---------|
| TensorNOODL | 2/13 | 4/11 | 1/14 | 10/24 |
| Mairal '09 | 1/10 | Not Found | 8/17 | 3/7 |
| Fu et al. [7] | 4/16 | 3/15 | 3/30† | 5/12 |

†The authors set the number of cluster to 5, here we combine the two clusters corresponding to "Executive".

**Figure E.1:** Enron Email Analysis. The plot and the table show the recovered group email activity patterns over time, and the cluster quality analysis, respectively. Note the increased legal team activity before the crisis broke out internally (Oct. '00), to public (Oct '01), till lay-offs.

to compress its dynamic range. Further, we also scale all elements by the magnitude of the largest element and subtract the mean (over the temporal aspect) from the non-zero fibers.

We initialization the dictionary using the algorithm presented in [1] for `TensorNOODL` and `Mairal '09`. This yielded 4 dictionary elements. As in case of the synthetic experiments, we set $\eta_x = 0.2$, $\tau = 0.1$ and $C = 1$. We set the dictionary update step-size $\eta_A = 10$, and run `TensorNOODL` in batch setting for 100 iterations. We recover the sparse factors $\mathbf{B}^{*(t)}$ and $\mathbf{C}^{*(t)}$ using our untangling Alg. 2. To compile the results, we ignore the entries with magnitude smaller than 5% of the largest entry in that sparse factor column.

**Evaluation Specifics**: As in case of [7], we use cluster purity (`False Positives/Cluster Size`) as the measure of the clustering performance. To this end, we also compare our results with [7]. Note that [7] solves a regularized least squares-based formulation for low-rank non-negative tensor factorization, wherein one of factor is sparse (corresponds to employees) and the others have controlled Frobenius norms. Here, the non-zero entries of the sparse factor gives insights into the employees who exhibit similar behavior. Unlike `TensorNOODL` and `Mairal '09`, this procedure however does not recover the email patterns of interest.

(a)

(b)

**Figure E.2:** Comparing the recovered signatures. Panel (a) shows the recovered signatures by `TensorNOODL` and (b) shows those for `Mairal '09`.

**Discussion**: The results of the decomposition by `TensorNOODL` are shown in Fig. E.1. The Enron organizational structure has four main units, namely, 'Legal', 'Traders', 'Executives', and 'Pipeline', which coincides with the number of dictionary elements recovered by `TensorNOODL`. Specifically, as opposed to [7], which take the number of clusters to be found as an input, `TensorNOODL` leverages the model selection performed by initialization algorithms. Fig. E.2 shows the comparison between the recovered signatures by `TensorNOODL` and `Mairal '09`, respectively. Note that [7] does not recover the signatures and only focuses on clustering. On this front, along with recovering the email activity patterns, `TensorNOODL` is also superior in terms of the clustering purity as compared to other techniques as inferred from the False Positives to Cluster-size ratio (Fig. E.1). The email activity patterns show how different group activities changed as time unfolded. In line with Diesner and Carley [6], we observe that during the crisis the employees of different divisions indeed exhibited

cliquish behavior. These results illustrate that our model (and algorithm) can be used to study organizational behavior via their communication activity.

Note that here we use `TensorNOODL` in the batch setting, i.e., we reuse samples. This shows that empirically our algorithm can be used in the batch setting also, although our analysis applies to the online setting. We leave the analysis of the batch setting to future work.

### E.2.2   Analysis of the NBA Dataset

**Figure E.3:** Shot Patterns and Teams in the NBA dataset. Panels (a-i) to (g-i) show dictionary factor ($\mathbf{A}^{(T)}$) columns (elements) reshaped into a matrix to show different recovered shot patterns. Here, the 3-point line and the rim is indicated in black. Corresponding sparse factor ($\widehat{\mathbf{B}}$) representing the Teams are shown in panels (a-ii) to (g-ii).

The online nature of `TensorNOODL` makes it suitable for learning tasks where data arrives in a streaming fashion. In this application, we analyze the National Basketball Association (NBA) weekly shot patterns of high scoring players against different teams. In this online mining application, our aim is to tease apart the relationships between shot selection of different players against different teams. Here, our model enables us to cluster the players and the teams, in addition to recovering the shot patterns shared by them.

**Figure E.4:** Structured tensor of interest $\underline{\mathbf{Z}}^{(t)} \in \mathbb{R}^{n \times J \times K}$ for the shot pattern analysis of NBA data. There are 27 such tensors arriving every week of the season.

We form the NBA shot pattern dataset by collecting weekly shot patterns of players for each week (27 weeks) of the $2018 - 19$ regular season of the NBA league. Each of these tensors consists of the locations of all shots attempted by players (above $80^{\text{th}}$ percentile of the 497 active players, which gives us 100 high-scorers) against (30) opponent teams in a week of the $2018 - 19$ regular season of the NBA league. To form the tensor we divide the half court into $10 \times 12$ blocks, and sum all the shots from a block to compile the shot pattern. We then vectorize this 2-D shot pattern, which constitutes a fiber of the tensor. Since players don't play every other team in a week, the resulting weekly shot pattern tensor $\underline{\mathbf{Z}}^{(t)} \in \mathbb{R}^{100 \times 30 \times 120}$ has only a few non-zero fibers, and fits the model of interest shown in

**Players corresponding to element 1**

| Players | Position | Coefficient Value |
|---|---|---|
| Harrison Barnes | Small forward / Power forward | -0.2770 |
| Stephen Curry | Point guard | -0.7620 |
| Kevin Durant | Small forward | -0.0707 |
| Nikola Jokic | Center | 0.5040 |
| CJ McCollum | Shooting guard | -0.0771 |
| Donovan Mitchell | Shooting guard | 0.0414 |
| Jamal Murray | Point guard / Shooting guard | -0.1677 |
| Jusuf Nurkic | Center | 0.0352 |
| Ricky Rubio | Point guard | 0.0191 |
| Klay Thompson | Shooting guard | -0.2128 |
| Russell Westbrook | Point guard | -0.0208 |
| Lou Williams | Shooting guard / Point guard | -0.0198 |

**Players corresponding to element 2**

| Players | Position | Coefficient Value |
|---|---|---|
| Harrison Barnes | Small forward / Power forward | -0.0187 |
| Danilo Gallinari | Power forward / Small forward | -0.0515 |
| Tobias Harris | Small forward / Power forward | -0.2729 |
| Donovan Mitchell | Shooting guard | 0.6536 |
| Karl-Anthony Towns | Center | 0.5449 |
| Andrew Wiggins | Shooting guard / Small forward | 0.4454 |

**Players corresponding to element 3**

| Players | Position | Coefficient Value |
|---|---|---|
| LaMarcus Aldridge | Power forward / Center | -0.2248 |
| Trevor Ariza | Small forward / Shooting guard | 0.3195 |
| DeMar DeRozan | Small forward / Shooting guard | -0.6716 |
| Bryn Forbes | Shooting guard / Point guard | 0.1241 |
| Justin Holiday | Shooting guard / Small forward | 0.1074 |
| Josh Richardson | Shooting guard / Small forward | 0.6049 |
| Justise Winslow | Point guard | -0.0580 |

**Players corresponding to element 4**

| Players | Position | Coefficient Value |
|---|---|---|
| Bojan Bogdanovic | Small forward | -0.0275 |
| Devin Booker | Shooting guard / Point guard | 0.0114 |
| Clint Capela | Center | -0.2256 |
| Willie Cauley-Stein | Center / Power forward | -0.0150 |
| Evan Fournier | Shooting guard / Small forward | 0.2032 |
| James Harden | Shooting guard / Point guard | 0.1992 |
| Buddy Hield | Shooting guard | -0.0198 |
| Jeremy Lamb | Shooting guard / Small forward | -0.1468 |
| Derrick Rose | Point guard | 0.4961 |
| Ricky Rubio | Point guard | 0.0198 |
| Pascal Siakam | Power forward | -0.0244 |
| Karl-Anthony Towns | Center | 0.7711 |
| Kemba Walker | Point guard | 0.0331 |
| Andrew Wiggins | Shooting guard / Small forward | -0.0119 |
| Thaddeus Young | Power forward | -0.0148 |
| Trae Young | Point guard | 0.0415 |

**Players corresponding to element 5**

| Players | Position | Coefficient Value |
|---|---|---|
| Devin Booker | Shooting guard / Point guard | 0.0104 |
| Clint Capela | Center | 0.0210 |
| Luka Doncic | Guard / Small forward | -0.0162 |
| Eric Gordon | Shooting guard / Small forward | 0.0150 |
| James Harden | Shooting guard / Point guard | 0.0678 |
| Tobias Harris | Small forward / Power forward | -0.0247 |
| Joe Ingles | Small forward | 0.1005 |
| Josh Jackson | Small forward / Shooting guard | -0.0100 |
| Donovan Mitchell | Shooting guard | 0.0984 |
| Kelly Oubre Jr. | Small forward / Shooting guard | -0.0143 |
| Derrick Rose | Point guard | 0.6507 |
| Ricky Rubio | Point guard | 0.0488 |
| Karl-Anthony Towns | Center | 0.6924 |
| Kemba Walker | Point guard | 0.1670 |
| Andrew Wiggins | Shooting guard / Small forward | 0.2000 |
| Lou Williams | Shooting guard / Point guard | 0.0196 |

**Players corresponding to element 6**

| Players | Position | Coefficient Value |
|---|---|---|
| Deandre Ayton | Center / Power forward | 0.0640 |
| Eric Bledsoe | Point guard | 0.0527 |
| Bojan Bogdanovic | Small forward | -0.1353 |
| Devin Booker | Shooting guard / Point guard | 0.4668 |
| Jimmy Butler | Shooting guard / Small forward | -0.0157 |
| Kentavious Caldwell-Pope | Shooting guard | 0.0507 |
| Clint Capela | Center | 0.6348 |
| Willie Cauley-Stein | Center / Power forward | -0.0303 |
| Jordan Clarkson | Point guard / Shooting guard | -0.0141 |
| John Collins | Power forward | 0.0948 |
| DeAaron Fox | Point guard | 0.0148 |
| Aaron Gordon | Power forward / Small forward | 0.0978 |
| Eric Gordon | Shooting guard / Small forward | 0.1861 |
| James Harden | Shooting guard / Point guard | 0.2834 |
| Buddy Hield | Shooting guard | -0.0135 |
| Justin Holiday | Shooting guard / Small forward | 0.0756 |
| Josh Jackson | Small forward / Shooting guard | 0.0339 |
| LeBron James | Small forward / Power forward | -0.1362 |
| Kyle Kuzma | Power forward | -0.0272 |

**Players corresponding to element 6 continued ...**

| Players | Position | Coefficient Value |
|---|---|---|
| Jeremy Lamb | Shooting guard / Small forward | -0.0229 |
| Kawhi Leonard | Small forward | -0.0384 |
| Brook Lopez | Center | 0.0194 |
| Lauri Markkanen | Power forward / Center | 0.0186 |
| CJ McCollum | Shooting guard | 0.0148 |
| Khris Middleton | Shooting guard / Small forward | 0.0617 |
| Jusuf Nurkic | Center | 0.0121 |
| Cedi Osman | Small forward / Shooting guard | -0.0260 |
| Kelly Oubre Jr. | Small forward / Shooting guard | -0.1673 |
| JJ Redick | Shooting guard | -0.0474 |
| Terrence Ross | Small forward / Shooting guard | 0.0216 |
| Pascal Siakam | Power forward | -0.0512 |
| Ben Simmons | Point guard / Forward | -0.0166 |
| Myles Turner | Center | -0.3469 |
| Nikola Vucevic | Center | 0.0827 |
| Thaddeus Young | Power forward | -0.0494 |
| Trae Young | Point guard | -0.1377 |

**Players corresponding to element 7**

| Players | Position | Coefficient Value |
|---|---|---|
| Harrison Barnes | Small forward / Power forward | 0.0330 |
| Mike Conley | Point guard | 0.2633 |
| Jae Crowder | Small forward | 0.0454 |
| Stephen Curry | Point guard | 0.0429 |
| Anthony Davis | Power forward / Center | -0.3173 |
| Luka Doncic | Guard / Small forward | -0.0239 |
| Kevin Durant | Small forward | -0.5214 |
| Marc Gasol | Center | 0.0655 |
| Paul George | Small forward | -0.6895 |

**Players corresponding to element 7 continued...**

| Players | Position | Coefficient Value |
|---|---|---|
| Jerami Grant | Forward | -0.0767 |
| Joe Harris | Shooting guard / Small forward | -0.0120 |
| Jrue Holiday | Point guard / Shooting guard | -0.2258 |
| Kyrie Irving | Point guard | -0.0128 |
| Julius Randle | Power forward / Center | -0.0266 |
| DAngelo Russell | Point guard | -0.0365 |
| Dennis Schroder | Point guard / Shooting guard | 0.1013 |
| Klay Thompson | Shooting guard | 0.0322 |
| Dwyane Wade | Shooting guard | 0.0208 |
| Justise Winslow | Point guard | 0.0431 |

**Table E.5:** Analysis of Sparse factor corresponding to Players ($\widehat{\mathbf{C}}$)

Fig. 1. In case a player plays against a team more than once a week, we average the shot patterns to form the weekly shot pattern tensor.

**Data Preparation and Parameters**: To prepare the data, we element-wise transform each non-zero element of the weekly shot pattern tensor $(\underline{\mathbf{Z}}^{(t)}(i,j,k))$ as $\underline{\mathbf{Z}}^{(t)}(i,j,k) = \log_2(\underline{\mathbf{Z}}^{(t)}(i,j,k)) + 1$ to reduce its dynamic range. We then subtract the mean along the shot pattern axis to reduce the effect of any dominant shot locations.

We form the initial estimate of the incoherent dictionary factor $(\mathbf{A}^*)$ from the $2017 - 18$ regular season data of the top $80^{\text{th}}$ percentile players using the initialization algorithm presented in [1]. We use $\eta_x = 0.1$, $\tau = 0.2$, $C = 1$ and $\eta_A = 10$ as the `TensorNOODL` parameters to analyze the data.

**Evaluation Specifics**: We focus on the games in the week 10 of the $2018 - 19$ regular season to illustrate the application of `TensorNOODL` for this sports analytics task. Our analysis yields the shared shot selection structure of different players and teams.

**Discussion**: In the main paper, we analyze the similarity between two players – James Harden and Devin Booker – who incidentally at that time were seen as having similar styles [10, 13]. In this case, our results corroborate that the shot selection patterns of these two players is indeed similar. This is indicated by sparse factor corresponding to the players.

In Fig. E.3, and Table. E.5 we show the recovered dictionary elements$(\mathbf{A}^{(T)})$ or the shot patterns and the corresponding clustering of teams $(\widehat{\mathbf{B}}^{(T)})$, and the players $(\widehat{\mathbf{C}}^{(T)})$, respectively, for week 10. For both $\widehat{\mathbf{B}}^{(T)}$ and $\widehat{\mathbf{C}}^{(T)}$ we show the elements whose corresponding magnitude is greater than $10^{-2}$. These preliminary results motivate further exploration of `TensorNOODL` for sports analytics applications. The theoretical guarantees coupled with its amenability in highly distributed online processing, makes `TensorNOODL` especially suitable for such application, where we can learn and make decisions on-the-fly.

## Footnotes

[1]They conjecture that the sample complexity is similar to the *biased* counterpart (`Arora(b)`), but leave the exact analysis to future work; see [1], Theorem 2, and footnote 2. In our experiments,