[Reviews · NeurIPS 2020]

Review 1

Summary and Contributions: The paper considers the task of online tensor decomposition, where the latent factors along one mode satisfy incoherence conditions, and the latent factors along the other modes are sparse. At each time step, a sparse set of columns in the tensor are fully observed. The column latent factors are assumed to be incoherent and fixed over time, while the other two modes are associated to sparse latent factors, and can vary over different time steps. By concatenating the dense columns of the observed tensor into a matrix, then the task of recovering the latent factors reduces to a dictionary learning problem. As a result, the proposed algorithm build off of previous dictionary learning algorithms, but adds an untangling step to recover the sparse latent factors of the tensor, and updates the dictionary across time steps via gradient descent. The theoretical result assumes incoherence of the column latent factors, as well as generative distributional assumptions on the sparse latent factors. The authors prove exact recovery with linear convergence to the latent factors. Finally they provide numerical simulations on synthetic data and NBA shot pattern data and verify empirically the linear convergence of their proposed algorithm.

Strengths: The paper is interesting and addresses the setting when different assumptions are placed on the latent factors associated to different modes of the tensor (in particular one mode being associated to incoherent latent factors, and the other two being sparse). Furthermore the paper considers an online setting rather than the typical batch setting for tensor decomposition. Their algorithm is amenable to fast and scalable implementation, and shows promise for real-world practicality from simulations. The linear convergence results for exact recovery also seem sound.

Weaknesses: The distributional assumptions as authors mentioned is a limitation of the model, but this is a good start and removing this assumption can be left for future work.

Correctness: Results seem correct as far as I can tell.

Clarity: Paper is well written.

Relation to Prior Work: Yes, the heterogeneously structured assumptions along with the online data streaming model are different from the models previously assumed in literature.

Reproducibility: Yes

Additional Feedback:


Review 2

Summary and Contributions: This manuscript studies a very specific streaming tensor decomposition problem and presents and analyzes it by framing the problem as a dictionary learning problem (allowable because two of the tensor factors are suitably sparse). Once cast as a dictionary learning algorithm the authors leverage existing algorithms to solve that problem.

Strengths: Because of the structure of the problem, the authors are able to build provable guarantees for their algorithm and the numerical results suggest it works well on synthetic examples relative to other work. The paper is also reasonable well presented and clear about the problem set up and algorithm.

Weaknesses: My main reservation with this paper is that, bluntly, it feels as though the model and assumptions are way too restrictive and have been chosen to allow the problem to reduce to that in [11]. This is exacerbated by limited development of the model. The algorithm is even essentially the same and I do not feel that the "disentangling" of B and C via the Khatri-Rao product is a novelty, these structural observations are quite common for tensor unfoldings. Moreover, there is a sense in which this is not really a tensor problem. It would seem equally fair to cast it as a dictionary learning problem with a specific distribution on the coefficients (induced by the choices for B and C here). Perhaps I am missing a novelty here, but the paper feels too close to [11] without suitably explaining if there is anything particularly interesting or novel added to the problem by casting it as a tensor one. - The authors seemingly make a big deal of the restrictive assumptions of other methods and the need for algorithms that work for more "heterogeneous" structures, but the structure here is quite restrictive and seemingly necessary given the algorithmic structure (because of the "reduction" to a dictionary learning algorithm). It is not clear to me the authors have made a sufficiently good case for the structural requirements they making being reasonable. For example, needing sparse but not too sparse B and C, or requiring that every entry in B and C can be non-zero, etc. - Related to the prior point, while they may not "provable" it would be interesting to see comparisons with the other common methods mentioned. Also, how effective is the method as the assumptions are relaxed in a synthetic problem (does it break down as soon as they are violated)? The "real" example shows the method does something, but it is not clear it adds much — maybe a more clear description of why this is a good model for that problem would help make the case. - Similarly, the authors note choices of model parameters (such as m) as a drawback of other methods, but doesn't requiring A^(0) be close to A^* in a very concrete sense mean this work requires knowledge of m as well? - (1) almost seems a bit misleading as everything after that point is written without fixed B and C, but "samples" Z^(t). Given a fixed tensor (say with sparse B and C) would the algorithm be applicable? Presumably, but that is a bit hard to extract since the main theoretical results only find A (converging in t) and the assumptions on B^(t) and C^(t) are quite different from them being fixed. Maybe there is a clear way to address this problem, in which case the authors should note it, but it does make the title seem a bit off as this is not quite a generic CP/PARAFAC decomposition even if A is incoherent and B and C are sparse.

Correctness: Since the bulk of the theoretical results are from [11], they are at least in alignment with that work.

Clarity: I think that the paper is reasonably easy to follow. One part that I feel could be improved (this ties into my points above) is the "model justification" section. I think a bit more concreteness would help along with a discussion of what happens if the model is misspecified.

Relation to Prior Work: In one sense this part is fine, though I do think that other tensor decomposition algorithms that are quite effective in practice are written off a bit too hastily and as overly restrictive (see my comments above), particularly in contrast to this work. Bluntly, I do not think the authors do a good job distinguishing between more general algorithms that may give up provably guarantees and algorithms that may be provably in certain very specific settings but perform poorly if the model is misspecified. (I am not necessarily saying this algorithm falls in the second category, but there is also no discussion of its robustness).

Reproducibility: Yes

Additional Feedback: - In algorithm a should Z^(t) be generated from (3) and not (1)? (1) does not have the "sampling" part of the model. - The broader impacts statement is *not* a place to profess significance of the work. It is a place to discuss potential societal implications (positive and negative). The authors should rework their statement to be in line with the stated goals for its inclusion. (It is okay if there is not a lot to say based on the paper contents, but what is said should be well thought out.) UPDATE: After reading the authors feedback and seeing the other reviews I have decided to raise my score as my primary concerns about the model formulation/assumptions were partially assuaged. I would still encourage the authors to better motivate the model and assumptions in the text upon revision and more clearly articulate how the assumptions are necessary (or not) for provable guarantees. In regards to robustness, the authors do explore robustness with respect to model parameters, but my inquiry was more directed at exploring the robustness to a misspecification in the model itself. I still think this would be an interesting line of inquiry.


Review 3

Summary and Contributions: This paper considers the problem of online structured CP decomposition. The model considered consists of a sequence of observed tensors Z_t = [[A,B_t,C_t]] having a low rank CP structure where the first factor A is shared across all time steps while the two other factors are independent random sparse tensors following some distribution (for each factor, the distribution is the same at all time steps). The authors provide a provable recovery algorithm which reduces the problem to a structured matrix factorization one which can be tackled efficiently using algorithms for dictionary recovery from sparse measurements. The theoretical guarantees provided are strong (exact recovery and linear convergence) under some reasonable assumptions, including incoherence of the factor matrix A and sparsity of the factor matrices B_t and C_t. The overall approach proposed by the authors is conceptually simple, as it simply reduces the tensor problem to a matrix one for which extensive literature and algorithms are available. To some extent the approach ignore the tensor structure of the problem (except for the recovery of the factors B and C from the Kronecker structured matrix X), which can be seen as a small caveat of the method (which is largely mitigated by the strong guarantees that are derived for this approach), I wonder whether more explicitly leveraging the tensor structure could lead to improved convergence time. Experiments on synthetic and real world data are presented. In the synthetic experiment, the proposed approach is compared with other classical dictionary learning / sparse coding algorithms. The experiment on real data (NBA shot pattern) only provides qualitative results for the proposed approach and no other method are compared to on this dataset. An additional experiment on real data is provided in the supplementary material (Enron dataset) where the evaluation is again mainly qualitative (comparisons with sparse coding and a tensor based approach [7] are provided).

Strengths: - The paper is well written - The approach proposed is conceptually simple and comes with strong guarantees

Weaknesses: - As mentioned by the authors the proposed approach is "agnostic" to the tensor structure of the problem, which as I mentioned above can be seen as a caveat/limitation of the proposed approach. - The experiment section is a bit limited, mainly wrt to few alternative approaches that are compared to and the qualitative nature of the NBA results which is in my opinion is nice to showcase the interpretable nature of the solution of the problems (but this has more to do with sparse coding in general) but fails to showcase how the method compare to alternative optimization algorithms. In particular, I think comparison with simple tensor based algorithms (e.g. ALS with l1 regularization on the two factors B and C) would greatly improve this section. Nonetheless, I think the quality of the theoretical analysis (though I did not carefully checked the proofs) and theoretical guarantees outweighs the fact that the experimental section is a bit limited. If possible, I would recommend the authors to include comparison with simple tensor based methods for the experiments as well as providing quantitative results on the real world datasets in addition to the qualitative ones.

Correctness: I did not carefully checked the proofs (10 pages in appendix) but the results seem sound.

Clarity: Yes, the paper is clearly written.

Relation to Prior Work: Yes the relation to prior work is clearly discussed though approaches from the tensor literature should be included in the experiments if possible.

Reproducibility: Yes

Additional Feedback: - In section 5.1: "Experimental set-up: We compare TensorNOODL with online dictionary learning algo- rithms presented in [6] (Arora(b) (incurs bias) and Arora(u) (**claim no bias**)), and [31]," I am a bit confused by the "claim no bias" in parenthesis. Does this means that the original paper claim that the approach incur no bias but that the authors of the present paper disagree with this? If it is the case, I would advise to either substantiate this claim or to remove the word "claim". It is also totally possible that I misunderstood this statement, in which case please clarify what is meant. - How is A^0 initialized for the experiments on real data? I think this may deserve some clarification. Also, in the synthetic experiments A^0 is initialized by perturbing the ground truth matrix A^*. Why is it the case and wouldn't it be more realistic to use one of the effective initialization techniques mentioned previously in the paper?


Review 4

Summary and Contributions: In this paper, the authors propose a provable online algorithm for structured CP tensor factorization. Bothe theoretical anlaysis and numerical experiments are presented.

Strengths: 1. It is an interesting topic. 2. The organization is good. 3. The analysis is appreciated.

Weaknesses: 1. For example in Table 1, such a result may not be that big a difference, since there are many existing works. A key contribution for a theoretical work may be some key equalities/inequalities are introduced, so that you may be able to conque some diffifculty of existing frameworks. 2. If the current Dl-based algorithm can converge in linear ratte, and checking your proof structure, a FISTA-based schme will lead to a quadratic rate, right? Please clarify it. 3. Is the initialization process necessary? In numerical experiments, I believe it can be removed. How about your theoretical analysis? Many current analysis frameworks have removed the initialization process. As for CP with scaling and permutation, the landscape is symmetric, it is very possible to achieve global optimal results, as the saddle points are strict and thus are easy to escape.

Correctness: Seems to be, did not check each formula.

Clarity: Yes.

Relation to Prior Work: Yes.

Reproducibility: Yes

Additional Feedback: After rebuttal, The depth os this work is appreciated. The authors' clarification also helps to fully address some concerns.

[Author Response · NeurIPS 2020]

We would like to thank all the reviewers for their comprehensive reviews. We clarify the major comments below.

**Robustness, Novelty [R2]** TensorNOODL is a simple online tensor factorization algorithm which provably recovers
the constituent factors exactly (upto scalings and permutations) at a linear rate for an inherently non-convex problem.
Our main result only states the *sufficient conditions*, and as shown in Fig. 4, the algorithm succeeds for all values of
$\{J, K, m, \alpha, \beta\}$ showing its robustness; see also App.E where it shows orders of magnitude superior performance as
compared to related techniques. In addition, our analysis of the resulting Khatri-Rao structure is, to the best of our
knowledge, also the first to establish guarantees for recovery of constituent factors from the Kronecker structured data.

**Model Assumptions [R1, R2]** For probabilistic analysis like ours, it is common to impose structure on the factors,
such assumptions are also used by existing provable algorithms for tensor factorization [1,2,3]. As discussed in Sec. 1.3,
the existing provable algorithms for tensor factorization either only apply to the undercomplete case ($m \leq n$) [7,8,9], or
when applicable for overcomplete case, use sum of squares [6], or tensor power iterations with multiple initializations
[1,3], which are cumbersome for practical use. As noted in Sec.6 (and suggested by **R1,R3**) this is a first of its kind
simple provable algorithm, and we aim to work towards relaxing the conditions in our future work. Note that although
we leverage results from [11], our analysis works with any dictionary learning algorithm with appropriate guarantees.

**Model Selection and Connections to popular algorithms [R2]** While in general, the model selection for tensor
factorization task is NP-hard [5] and is usually based on heuristics, for this particular structured tensor factorization
task, our model assumptions ensure that the initialization algorithm such as [2] can recover $\mathbf{A}^{(0)}$ as required by
A.2., implicitly accomplishing the model selection (finding $m$). Further, our aim here is to establish the theoretical
underpinnings behind popular heuristics (such as alternating optimization and exploiting the Khatri-Rao structure)
employed in tensor factorization tasks to enable development of simple, provable, and practical algorithms. Here, (1)
defines a general CP/PARAFAC problem only. As discussed in Sec.1, 1.1, 2-4, and Fig. 1, TensorNOODL accomplishes
*online* CP decomposition of a *structured tensor*, useful in applications where batch processing is not viable.

**Comparisons with ALS and using Tensor structure [R3]** We compare TensorNOODL with related techniques which are also agnostic to the tensor structure for a fairness. Out of these, Mairal '09 [4] can be viewed as a variant of $\ell_1$-based Alternating Least Squares (ALS). In addition, our result also shows that for this task, where a number of mode-1 fibers are zero, processing only the non-zero fibers may lead to significant gains since there is no need to solve large sparse approximation subproblems as is the case with ALS (which also need to be tuned). Therefore, it seems that leveraging tensor structure may increase the computational complexity. Nevertheless, it can be potentially be useful in presence of noise, where this structure is not obvious. Thank you for this insight.

Fig.A Comparing signatures of TensorNOODL & [4].

**Arora '15["unbiased"] and Qualitative Real-data Experiments [R3]** The authors propose an *unbiased* counterpart
in [2]. They conjecture that the sample complexity is similar to the *biased* counterpart, but leave the exact analysis
to future work; see [2], Thm 2, and footnote 2. In our experiments, we find that given equal number of samples, the
unbiased version is only slightly better than the biased counterpart. For quantitative real-world data, we compare the
cluster purity of TensorNOODL with other techniques [4,12] in App.E.1 for the Enron dataset. We've added Fig.A
which shows the qualitative signatures recovered, note that [12] does not recover the signatures.

**Initializing $\mathbf{A}^{(0)}$, Random initializations [R3,R4]** For real data experiments we use the algorithm proposed by [2] to
initialize the algorithm (line 261, App. E.2.). For synthetic data experiments, the initialization algorithm of [2] with a
running time of $\tilde{\mathcal{O}}(m^2 n^2 s)$ becomes the bottleneck (in terms of time) as $m$ and $s$ increase. Therefore, we perturb the
ground-truth dictionary (in accordance with our theoretical result) for these experiments, providing all methods with this
initialization. Further, TensorNOODL requires the initial dictionary estimate to follow A.2. for exact recovery at a linear
rate. Initializations which do not meet these conditions may still converge, albeit not at a linear rate. Empirically, we do
see evidence in support of this, and we agree with R4 that using random initializations (along with model selection
algorithm) is a very interesting direction for our future work.

**FISTA [R4]** TensorNOODL at its core uses Iterative Hard Thresholding (IHT) for sparse approximation, which
essentially solves a projected gradient descent problem for the $\ell_0$ problem (counterpart of ISTA which operates for the
$\ell_1$ case). Indeed, it is possible to build upon our work to use other algorithms for faster convergence. In theory, any
algorithm which can preserve the signed-support recovery of $\mathbf{x}$ can be used; see Lem. 3.

[1] Anandkumar, Ge, Janzamin ('15). Learning overcomplete latent variable models through tensor methods. (COLT). [2] Arora, Ge, Ma, Moitra ('15). Simple, efficient,
and neural algorithms for sparse coding. (COLT). [3] Sun, Lu, Liu, Cheng ('17). Provable sparse tensor decomposition. (JRSS). [4] Mairal, Bach, Ponce, Sapiro ('09).
Online dictionary learning for sparse coding. (ICML). [5] H astad ('90). Tensor rank is np-complete. (JoA) [6] Tang ('15). Guaranteed tensor decomposition: A moment
approach. (ICML) [7] Anandkumar, Ge, Hsu, Kakade, Telgarsky ('14). Tensor decompositions for learning latent variable models. (JMLR) [8] Anandkumar, Jain, Shi,
Niranjan ('16). Tensor vs. matrix methods: Robust tensor decomposition under block sparse perturbations. (AISTATS) [9] Sharan and Valiant ('17). Orthogonalized als:
A theoretically principled tensor decomposition algorithm for practical use. (ICML) [11] Rambhatla, Li, Haupt ('19). NOODL: Provable Online Dictionary Learning
and Sparse Coding. (ICLR) [12] Fu, Huang, Ma, Sidiropoulos, Bro ('15). Joint tensor factorization and outlying slab suppression with applications. (TSP)


[Meta-Review · NeurIPS 2020]

The paper received four positive reviews. After discussion, the reviewers seem satisfied with the rebuttal. The area chair agrees with their assessment and follows their recommendation.